# The Role of Stakeholders in the Context of Responsible Innovation: A Meta-Synthesis

**Luciana Maines da Silva** [1,*] **, Claudia Cristina Bitencourt** [1] **, Kadígia Faccin** [1] **and Tatiana Iakovleva** [2]

1. Unisinos Business School, Unisinos University, Porto Alegre 90.470-280, Brazil; claudiacb@unisinos.br (C.C.B.); kadigiaf@unisinos.br (K.F.)
2. UiS Business School, Stavanger University, 8600 Stavanger, Norway; tatiana.a.iakovleva@uis.no
* Correspondence: lucianamaines@unisinos.br; Tel.: +55-519-997-204-05

**Abstract:** This paper contributes to the sustainability debate by analyzing the inclusion dimension in the responsible research and innovation (RRI) process. RRI is claimed to be an important tool for addressing global challenges and achieving sustainable development goals. While stakeholder involvement is considered to be imperative for the RRI process, there is little empirical evidence on (1) who the stakeholders participating in the RRI process are; (2) when stakeholders participate; (3) how stakeholders' inclusion contributes to the sustainable innovation process; and (4) who the agents are who orchestrate stakeholders' inclusion. This paper addresses the issue of stakeholder involvement through the lens of innovation management literature by attempting to link the innovation process to the responsibility concept. We employed a meta-synthesis of empirical studies of RRI to develop a deep understanding of stakeholder inclusion. After screening 139 articles, we identified seven empirical papers highlighting RRI process, mainly from projects nested in academic contexts. The findings indicate that multiple stakeholders are included at a late stage of the innovation process—during the market launch. To some extent, this allows for the adaptation of the solution, but such adaptations are limited in nature. This study also identifies the agents who stimulate stakeholder inclusion as being mainly academic researchers and researchers linked to multi-institutional projects. Our findings indicate that innovation management thinking is rarely applied in the governance of research and innovation projects 'born' in academia. We suggest enhancing RRI theoretical development by incorporating elements of innovation management such as early inclusion of users in the innovation process. For practitioners, this means an extension of the design space to allow early stakeholder inclusion in the innovation process to ensure responsible outcomes. We also identified avenues for future research. There is a need to systematically investigate which tools and frameworks for deliberate stakeholder inclusion are relevant at the various stages of the innovation and development process.

**Keywords:** responsible research and innovation; responsible innovation; inclusion; stakeholder; stakeholder inclusion; innovation process; sustainable development goals; meta-synthesis

---

## 1. Introduction

Seeking to transform the world, United Nations has developed 17 Sustainable Development Goals (SDGs), that is, "a plan of action for people, planet and prosperity" [1]. Those goals intend, by 2035, to stimulate action in critical areas including poverty and hunger (people); the degradation of natural resources and climate change (planet); economic, social and technological progress (prosperity); and peaceful, just, and inclusive societies (peace).

SDG involves governments, which can act through policies and regulations, even though these policies and regulations are still in the development phase [2]. It also concerns private initiatives, aligning research and development (R&D) demands with societal values, needs, and expectations [3]. The Grand Challenges transcend national borders and have effects on large numbers of people, communities, and the planet as a whole, thus demanding a collaborative effort to find solutions [4]. Responsible research and innovation (RRI) is suggested as a method to govern innovation development to address challenges such as poverty, inequality, aging population and availability of quality healthcare [5,6]. Such principles suggest broader stakeholder inclusion in the decision-making process, anticipation of societal needs, and reflection of concerns [7], which calls for new innovation policies [8]. Stakeholder inclusion is heavily connected to SDG17, which seeks to "strengthen the means of implementation and revitalize the Global Partnership for Sustainable Development". RRI can become a means for highlighting multi-stakeholder partnerships to "mobilize and share knowledge, expertise, technology and financial resources" (p. 27 [1]) and at the same time to "encourage and promote effective public, public-private and civil society partnerships, building on the experience and resourcing strategies of partnerships" (p. 28 [1]).

The term RRI emerged as a political agenda in 2002 in the 6th EU Framework Program for Research and Technological Development (FP6), which is a set of actions at EU level to finance and promote research [9]. Recently, RRI was promoted by the European Commission through European Framework Programmes for Research and Innovation, "FP7" and "Horizon 2020". RRI aims to "ensure desirable effects of technology and capture a high level of responsibility in R&I initiatives" (p. 216 [10]). RRI is often performed from a policy or socio-ethical perspective and focused on academic R&D environments. Although, most innovations occur in commercial or industrial settings [11,12]. Thus, responsible innovation (RI) has emerged as a concept close to RRI, but it has a fine-grained focus on the innovation process itself [12–14]. For the purpose of this paper, we will use the term RRI. Whilst this will include RI, we will in particular keep our focus on management of innovation process with regards to stakeholder inclusion.

Considering innovation as a process suggests a guideline for anticipation, inclusion, responsiveness and reflexivity, as well as knowledge management [14]. These dimensions imply a collective, ongoing commitment to evaluate potential positive and negative consequences of research and innovation in dialogue with a broad range of stakeholders and through a reflective process that delivers responsible outcomes. The inclusion dimension (or deliberation) resonates in almost all RRI studies [14], since it addresses the inclusion of stakeholders in the research and innovation process. Owen et al. [15] and Stilgoe et al. [16] use the terms inclusion and deliberation indistinctly. Stakeholder inclusion focuses on who to involve, during which stage of the innovation process, and whether the stakeholder network is representative. Deliberation focuses on the decision-making process [14]. Inclusion has the objective of broadening visions, purposes, issues, and dilemmas for wide and collective deliberation through the processes of dialogue, engagement, and debate [15]. Inclusion intends to develop greater democratic accountability in the innovation life cycle [17]. Von Schomberg (p. 1 [13]), argues that "RRI should be understood as a strategy of stakeholders to become mutually responsive to each other and anticipate research and innovation outcomes underpinning the 'grand challenges' of our time for which they share responsibility".

Despite the clear importance of stakeholders in RRI, empirical evidence in literature addressing their inclusion is still scarce [18]. RRI has stalled at the point of articulating a process of governance with a strongly normative loading, without clear practical guidelines toward implementation practices [19]. This has occurred in spite of general recognition of the importance of stakeholder inclusion, anticipation of their needs, and reflection on their feedback supporting decisions relating to new solutions offered to society.

Therefore, many questions remain about who, when, and how to include, anticipate, and reflect [14,20,21]. Stakeholder involvement is instrumental in order to achieve the SDGs. Therefore, the highlighted gaps in the knowledge base about stakeholder involvement into RRI are important to

recognize. In this article, we aim to understand the role of stakeholders in the RRI process. Specifically, we consider the following: (1) who are the agents who orchestrate stakeholders' inclusion; (2) when stakeholders participate—at which stage of the innovation process do they participate; (3) how stakeholders' inclusion contributes to anticipation and reflection on the innovation process—and what kind of innovation; and finally (4) who are the stakeholders typically invited to participate in the RRI process. In this way, we aim to bridge RRI as a normative concept to implementation practices, which might both drive theory development and provide practical applications for practitioners.

Innovation practices at the organizational level are well researched and documented in other domains, like innovation management literature. Within this literature, new models focusing on user inclusion in the innovation process are emerging. Examples of theories that describes this phenomena include, but are not limited to, agile stage-gate [22], and free innovation, developed by consumers and users who self-reward their efforts and do not charge for their projects [23]. The innovation management literature also emphasizes mechanisms and tools for stakeholder inclusion, like design thinking [24] or project management literature, like PMI Book [25]. Whilst this paper primarily seeks to contribute to RRI literature, it will utilize innovation management lenses to analyze the RRI as a process [26] in order to answer the above-mentioned research questions.

In the following sections, theoretical references informing the research are presented. This is followed by the method, discussion, and the conclusion.

## 2. Responsible Innovation

The debate on responsibility is not new and several fields have joined it over last thirty years [3]. Prominent among them have been the science technology studies (STS) [27,28] and corporate social responsibility (CSR) [29,30]. In STS, the concept of responsibility emerged from a post-war concern about the potential negative impacts of technologies like nuclear power. In the 1960s and onward the STS community became an influential force in the discussions. Recently, similar developments have occurred in innovation studies and the broader research environment emphasizing the importance of RRI. The CSR debate can contribute to innovation, good business practice, and performance by identifying business opportunities that explore new paths. These new paths contribute to face environmental and societal challenges by creating a comprehensive workplace in the supply chain responsive and committed to social and environmental issues and through creativity that leverages social practices (such as diversity and inclusion), promoting a common set of values [29]. Socially responsible innovations present an opportunity to create better innovations with lower implementation costs based on new forms of cooperation with different networks (alliances, cluster, virtual, etc.) [30]. However, CSR literature have recently been criticized, suggesting that CSR concept suffers from 'three curses'—that it is incremental, peripheral, and uneconomic [31]. It is therefore imperative to understand how all levels of society can be included in deliberative innovation processes, which offer solutions to the 'Grand Challenges' which society faces. Arguably, a radical transformation of CSR would involve principles like creativity, scalability, responsiveness, and locality [31], which are achievable via deliberate inclusion of diverse stakeholders and reflection upon their voices.

Responsible research and innovation evokes a collective duty of care: a commitment to rethinking the purposes and impacts of innovation as well as a reflection on how to make its pathways sensitive to uncertainty [32]. The concept of RRI is gaining importance because of its potential to contribute to the search for solution to challenges, which can guarantee sustainable development for a fairer society, achieved through integration [19]. This sustainable development can be achieved, for example, through deals with the poorer population, known as the base of the pyramid. In this way, companies contribute by improving lives, through the production and distribution of products and services in a culturally sensitive, environmentally sustainable, and cost-effective way [33].

These challenges refer to a greater perspective and quality of life [34]. This search for solutions demands the inclusion of stakeholders, which enables new visions, purposes, questions, and dilemmas to broaden the collective deliberation through processes of dialogue, engagement, and debate, inviting

and listening to broader perspectives of audiences and diverse stakeholders, and revolves around a quest for social legitimacy of innovation [35]. Involving stakeholders makes decision-making processes more open and participatory and more focused on sustainable development [19].

Inclusion is the most discussed dimension in RRI, but this does not mean that the theme has been exhausted. On the contrary, empirical evidence is still scarce [18]. Following Owen et al. [7] seminal article on RRI governance, several studies seek to understand the capacities for inclusion [36], the management practices [18], the stakeholders [13,16,37,38] and the difficulties and critical points in the inclusion process e.g., [18,20,38].

There is a tendency for technology actors to dominate social actors in responsibility processes, reducing the 'citizen' to one common actor because of the difficulties of translating messy social voices [19]. Through inclusion, anticipation, and reflection, it is possible to achieve a more balanced view, via new voices in the governance of science and innovation as part of a quest for legitimacy [16]. In this way, for example, users such as patients (in healthcare innovations) contribute to improving the final design, that will better meet their needs [39].

Inclusion is defined as the exchange of views between stakeholders, commonly agreed, based on shared information and evaluation criteria, which support stakeholders' decision-making on the innovation process and/or results [14]. This allows the introduction of a wide range of perspectives to reformulate issues and to identify potential contestation areas [15]. The recognition and knowledge of engagement beyond key stakeholders, diversifying inputs and delivering governance, opening up problem frameworks, recognizing engagement as a learning process, and opening up the discussion on future social perspectives are considered key indicators of inclusion [7].

By reflecting not only on the need for inclusion but also on how this inclusion occurs, important factors can be pointed out, such as stakeholders' opportunity to participate, or not, in research or innovation, the role of participants, and the power of relationships during inclusion [36]. The opportunity is linked to the choice to participate, considering that research and innovation should bring benefits to all stakeholders [38]. In many cases, some participants do not feel free to participate, especially as the theme may elicit ethical discussions on topics such as nanotechnology and biotechnology [36]. Otherwise, inclusion should be sensitive to the culture and the unique needs of the participants [40,41]. Another critique of inclusion is the failure of organizations to recognize the diversity of the public and institutions that can participate in the innovation process and governance of science, technology, and innovation [19]. Sometimes it is necessary to create spaces and educate the stakeholders about the subject in which they are involved [38].

Further critical points are excessive inclusion, which may jeopardize the integrity of the common good [21], as well as informational asymmetry [18]. The result of this mismanagement is the absence of a decision or an unresolved decision [42]. Following this point of view, we highlight the fundamental philosophical differences between actors and stakeholders (micro level); within the organizational structures of innovation systems (meso level); and relating to wider political, economic, cultural, and social contexts (macro level) [43]. But who are the stakeholders, and who invites them to participate (actors of inclusion)? The following sub-section will address these questions.

## 2.1. Stakeholders in RRI

Stakeholders are any group or individual that can affect or be affected by the fulfilment of the goals defined by the organization [28]. They may be classified as internal or external groups [44], at the same time, they may be classified as economic or non-economic actors [18]. Internal stakeholders are regarded as internal to the organization. The external group is composed of social and political actors who play a fundamental role in the credibility and acceptance of business activities, including governments, competitors, consumer advocates, environmentalists, special interest groups, and the media [36].

Extending this debate, RRI suggests the inclusion of external stakeholders, like individual researchers, research organizations, research ethics committees and their members, research and

innovation users, civil society of different levels with political decision-making powers, professional bodies, legislators, educational organizations, and public bodies [38]. Recent discussions have focused on how to engage lay citizens [45].

In addition, employees, users, supply chain stakeholders and external research institutes (universities and research centers) make important contributions [16]. Other classifications are suggested by Blok et al. [18], who consider economic (e.g., employees and suppliers) and non-economic (e.g., NGOs and research institutes) stakeholders. Furthermore, von Schomberg [13] proposes multi-stakeholder involvement, bringing together actors from industry, civil society, and research.

Despite all these classifications, the critical aspect is to define the stakeholders who need to (and can) be included and how they can contribute to innovation. The following section will discuss those points.

### 2.2. The 3W1H of Stakeholder Involvement—Exploring the Innovation Process

Innovation process theories typically describe a "development funnel" of innovation as sequential process consisting of several stages, including outlined concept, detailed design, testing, and launch phases [26]. Although recent literature has extended far beyond the classical "technology-push" model of innovation [46] and advanced our view on the innovation process through more flexible innovation models such as agile gate stage [22], lean start up [47], design thinking [24], project management [25], open innovation [48], and other techniques, it does not seem to be reflected upon in RRI literatures. On the other side, RRI and CSR studies tend to describe other techniques, sometimes overlapping with major innovation management practices, such as walkshop approach [49], engagement workshops [50–52], online platforms [53]; online knowledge sharing [54,55], social experimentation and design thinking [56,57], anticipation of risks and technology assessment [58,59].

Knowledge sharing, in innovation management literature and RRI literature, plays a crucial role in involvement of stakeholders. Innovators need to acknowledge the fact that knowledge within a particular community, would still be limited to address overall socioeconomic, environmental, and ethical issues in society. The effectiveness of stakeholder involvement can be determined by how efficiently complete and relevant information is obtained from all appropriate sources, transferred to (and processed by) those responsible and combined to generate a response [60] or reconfiguration of the initially proposed model. The themes within the RRI tools domain as well as innovation management are therefore highly concentrated on possible ways of accumulating knowledge and successfully deploying it in order to overcome societal and environmental challenges. While innovation management literature is focused on consumer and user involvement as well as the value chain for business, RRI literature emphasizes broader stakeholder inclusion with economic as well as non-economic stakeholders.

One approach to stakeholder inclusion is found in innovation management literature that addresses the needs and practices of innovation-oriented businesses that are close to the market and have commercialization and profit generation as their major goal [26]. Another approach is found in RRI literature's discussions on governance of research-based innovations within early stage research. In this case, the guidance aims to benefit society and addresses SDGs and grand challenges. Arguably, RRI literature is more concerned with research-based innovations, often situated in an academic context. Academic innovations are characterized by high levels of uncertainty both for solution and for market [61]. As demonstrated by the literature, in such cases there is a need to make the design space open for a much longer time to allow reflections and modification of the solution before the dominant design is chosen [62], thus stakeholder inclusion and reflection on their feedback becomes extremely important.

In this paper, we argue that it is worth applying lenses of innovation management and its vision of innovation process as a 'development funnel' to study stakeholder inclusion in RRI. Here we highlight four aspects, which lead to four questions. These questions are synthesized by the acronym '3W1H' (who (are the agents inviting stakeholders), when (the inclusion occurs), how (the inclusion occurs), and who (are the stakeholders)).

Stakeholder inclusion does not necessarily follow a pattern and may vary according to the nature and flow of information between those responsible for the exercises and the participants. Despite the clear importance of the stakeholder involvement in the innovation process, some aspects of inclusion are not yet sufficiently studied. In the context of RRI, there are no studies describing the agents within organizations or projects who are responsible for stakeholder inclusion. Recent studies identify a need to detect these agents, who will be responsible for stakeholder dialogue and integrating the knowledge generated into the company's processes [16]. The dialogue and knowledge integration are part of a major process that begins with the choice of who must participate. This process can be called "orchestration", which means the managerial action on resources [63–65]. This raises the following question: Who are the agents who orchestrate stakeholders' inclusion?

In addition, the development of products/services is a complex process that requires the management of several factors at different stages [63]. In general, it moves from concept to product marketing (market introduction) through project design and testing [63]. More complex processes can be understood using an agile stage gate model [22]. The model benefits are faster product launch, better response to user needs, better communication and team morale [22]. Agile stage-gate considers three stages: discovery & ideation, concept & business case, and development & launch. The model considers the stages common to development of new products, which are composed of a group of activities complemented by decision points (gates), or control points. Decision points can serve as opportunities for inclusion offering moments at which the process can be either continued or can be stopped (go/kill strategy). In RRI, inclusion should occur at the early stages of innovation [3], but, although some theoretical articles deal with the dimension of inclusion [3,7] (among others), none of them explore the theme of the innovation process. This leads to our next question: when stakeholders participate, at which stage of the innovation process does this happen?

Innovation can take several forms, summarized in four dimensions of change: product innovation (change in the products/services offered by a company), process innovation (change in the way in which the products/services are offered or presented to the consumer), position innovation (change in the context in which the products/services are introduced to the market), and paradigm innovation (change in the basic mental models that guide the actions of the company) [66,67]. Although it is related to all types of innovation, RRI research does not highlight the dimensions of change. Thus, we raise the third question: How does stakeholders' involvement contribute to innovation—and which kind of innovation?

Finally, we discuss the stakeholders who are involved in RRI. As shown in Section 2.1, many studies list the people who must be considered as stakeholders but do not take into consideration the responsible innovation context. The main purpose in the selection of stakeholders lies in the choice of who will actually contribute, but also those who will be impacted. While innovation management literature emphasizes user and customer inclusion, RRI literature advocates for broader inclusion of both economic and non-economic stakeholders. One persistent debate concerns appropriate inclusion [20], difficulties concerning conflicts of interest [67], fear of loss of power over the process [14], and fear about the relationship between secrecy and transparency [68], as well as operational aspects such as the consumption of time and other resources [37]. Accordingly, we present the fourth and last question: who are the stakeholders who should participate in the RRI process?

All those questions, which we named the 3W1H of inclusion, will guide the meta-synthesis described in the next section.

## 3. Meta-Synthesis Method

The aim of the study as alluded to earlier is to understand the role of stakeholders in the RRI process. This specifically consider the 3W1H: (1) who the agents are; (2) when stakeholders participate; (3) how stakeholders' inclusion contributes to innovation; and (4) who the stakeholders are. We applied the method of meta-synthesis, based on qualitative case studies, which produces a new and integrative interpretation of the findings that is more substantial than those resulting from individual investigations. This methodology allows the clarification of concepts and standards and results in

improvement of the existing knowledge states and emerging operational models and theories [69]. The focus was the analysis of the evidence in all the studies as well as the construction of theory, aiming to guarantee sensitivity to the contextual considerations of the primary studies [70]. Following the establishment of the research methodology, the remaining steps are described below.

### 3.1. Framing the Research Question

The first stage of the meta-synthesis, proposed by Hoon [70], concerns the conceptual framework of the theme, with the objective of identifying theoretical gaps that can be filled later. With this aim, we sought to deepen the concept of responsible innovation theoretically. The literature highlights the importance of the role of stakeholders in this process. In the light of the role of stakeholders in the RRI process, the first step was developed, and Section 2 presents the guiding questions for the further analysis. In this section, we present the procedure for the selection of empirical studies subjected to deeper analysis.

### 3.2. Finding Relevant Research

In this step, the articles that can be considered relevant for the meta-synthesis are identified. We decided to include only published works to ensure scientific quality of findings, although other methods of inclusion (scoping review including conference proceedings and book chapters for example) might have provided a broader set of literature. However, only peer reviewed articles are chosen for analysis in the present study. To locate a set of existing qualitative case studies, we performed a search of the databases of the Portal de Periódicos CAPES, EBSCOHost and Web of Science. The search was carried out on 7 September 2017. The search of the Portal de Periódicos CAPES was conducted using the exact terms "responsible innovation" AND "stakeholder"; "responsible research and innovation" AND "stakeholder"; and "responsible innovation" AND "stakeholder". The search generated 51 articles, which, after checking and excluding duplicates, was reduced to 36. The search of EBSCOHost involved selecting all the databases and using the terms "responsible innovation" OR "responsible research and innovation" AND "stakeholder", resulting in 146 articles, which, after checking and excluding duplicates, resulted in 72 articles. Finally, a search was performed of the Web of Science, including the terms "responsible innovation" OR "stakeholder", which produced 49 articles. When all the articles were listed, 157 articles were obtained initially. After checking and excluding duplicates between the databases, 139 articles remained. In addition, the non-indexed *Journal of Responsible Innovation* was included, due to its pertinence and relevance to the theme. The search of this journal used the keyword "stakeholder", which resulted in 78 articles.

In total, titles and abstracts of the 217 articles were analyzed (Please see the Supplementary Materials), identifying and excluding 16 as false positives; that is, they were not directly related to the theme. Of the excluded articles, 11 were from other areas that were not linked to research and responsible innovation (education, civil construction, and medical practices) and 3 referred to news, 1 to speech, and 1 to a calendar. The 217 articles were published between 2001 and September 2017. Figure 1 shows their distribution over time.

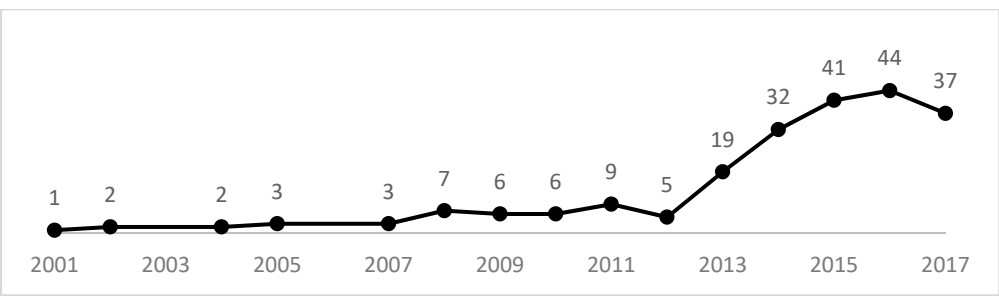

**Figure 1.** Number of articles published per year.

We can clearly see that the stakeholder issue, in RRI and RI, has been covered in an increasing number of articles in the last five years, confirming the importance of the issue. The relationship of stakeholders in the RRI context is consistent with other research, such as the study by Genus and Iskandarova [71]. The authors identify that most of the research on RRI has been published during the period 2011–2015, most of it being conducted in sector contexts concerning the dissemination of RRI tools and practices through funded projects (mainly the European Commission).

Table 1 presents the journals in which two or more articles were published. Since there is a specific journal for the topic, the *Journal of Responsible Innovation*, it is not surprising that it contains most papers. However, there are another 110 journals or calls for conference that show an interest in the topic in diverse areas, such as public policy, engineering, biotechnology, design, psychology, business, nanotechnology, and others. The full list is presented in Appendix A.

**Table 1.** List of publications.

| Total | Publication |
|---|---|
| 78 | *Journal of Responsible Innovation* |
| 8 | *Science & Public Policy* |
| 4 | *Science and Engineering Ethics* |
| 3 | *Asian Biotechnology & Development Review* |
| 3 | *Design Issues* |
| 3 | *Science & Engineering Ethics* |
| 3 | *Science and Public Policy* |
| 2 | *Bized* |
| 2 | *Cyberpsychology, Behavior & Social Networking* |
| 2 | *IEEE Technology and Society Magazine* |
| 2 | *Information Polity: The International Journal of Government & Democracy in the Information Age* |
| 2 | *International Conference on Research Challenges in Information Science* |
| 2 | *Journal of Business Ethics* |
| 2 | *Journal of Nanoparticle Research* |
| 2 | *Nanoethics* |
| 2 | *Research Policy* |
| 2 | *Social Sciences* |
| 2 | *Technological Forecasting and Social Change* |

In the sequence, we identified through the title, keywords, and abstract, the theoretical articles (conceptual and revisions). Hoon [70] points to a variation of content in abstracts and keywords, not indicating the research method used. When the same problem was encountered, it was necessary to obtain the full-text versions of many of the articles to categorize them manually. This process identified 104 conceptual articles, 10 editorials, 14 review studies, 12 quantitative studies (e.g., surveys), and 61 qualitative studies. Lastly, each of the 61 qualitative studies was classified, resulting in 3 ethnographic, 1 action research, 1 phenomenological, 1 field, and 55 case studies.

### 3.3. Inclusion/Exclusion Criteria

The next step in conducting the meta-synthesis was the appropriate inclusion of relevant qualitative case studies. The importance of specifying and applying the inclusion and exclusion criteria is to ensure the validity of the synthesis, which depends on the quality of the primary studies on which it is based [72].

The inclusion/exclusion criteria are presented in Table 2.

Following the predetermined criteria, of the 55 articles using case studies, 15 articles with illustrative cases were excluded; 15 articles in which the analysis does not correspond to the responsible innovation process or does not consider stakeholders were also excluded; 7 articles on cases based exclusively on documents (reports or policies), 8 case studies that use quantitative methods for the data analysis, 2 articles that are not available in the databases, and 1 that is a case of an RRI teaching method were also excluded. Finally, seven articles (Appendix B) remained, that according to the analysis

contribute to the understanding of the role of stakeholders in the context of responsible innovation. The seven articles are related to research projects. Blok [18] highlight the low number of empirical papers on RRI in business, and our findings confirm this notion. Figure 2 presents the flow diagram of the study selection.

**Table 2.** Inclusion/exclusion criteria.

| Criteria | Rationales | Reasons for Exclusion |
|---|---|---|
| 1. Qualitative case study | The meta-synthesis was restricted to articles that report qualitative case studies, ensuring that there is no difference between the research method that the primary researchers claim to have used and the approach actually used [17]. Also excluded were articles that use illustrative cases. | 24 articles were excluded. The cases were merely illustrative, being case studies with quantitative analysis or, moreover, cases that were a method of teaching about RRI. |
| 2. Access to the study | Some studies are not available in full, only the abstract being available. | 2 articles were excluded because they were not made available in full. |
| 3. Quality analysis | Quality criteria defined by Eisenhardt [43] and Yin [44], such as rigor, the link between theory and practice, the contextualization of the case and multiple data sources. | 7 articles that presented cases based exclusively on documents, such as reports or policies, were excluded. No articles were excluded due to a lack of quality. |
| 4. Study highlights the role of stakeholders in the analysis of the case(s) | Since the objective of the present study is to evaluate the inclusion of the stakeholders in the context of responsible innovation, to evaluate whether there is direct inclusion of the stakeholders in the research. | 15 articles were excluded because the cases presented did not refer to the innovation process or stakeholders in the case study. |

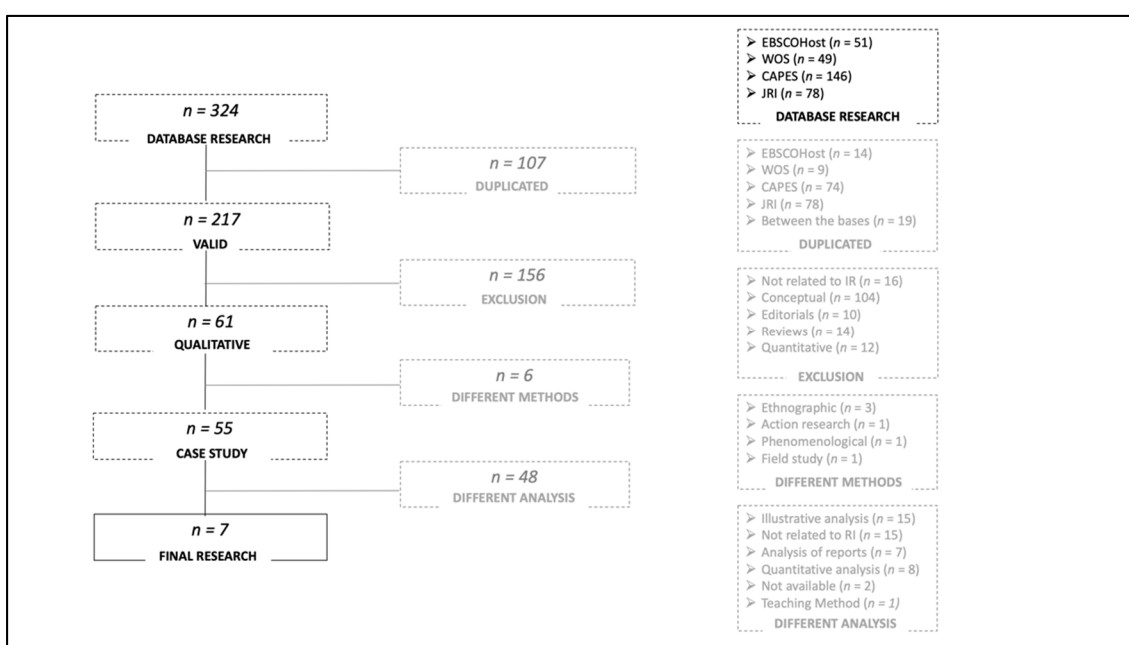

**Figure 2.** Flow diagram of the study selection.

The remaining seven articles were published between 2015 and 2017 in different countries and sectors. All the studies clearly describe the methods of analysis, applying research strategies that are consistent with best practices, following the recommendations of Eisenhardt [73] and Yin [74].

*3.4. Extracting and Encoding the Data*

The next step proposed is the extraction, coding, and categorization of the studies' evidence. The empirical material that serves as the basis for the meta-synthesis should highlight the insights

of the original researchers through the understanding and interpretation of the data by the meta-synthesis [70].

To obtain the necessary data, a reading guide (the result of the extraction will be available on Research Gate and available to reviewers in a separate file: "Additional Materials") (Table 3) was developed.

**Table 3.** Reading guide.

| Section | Observed Items |
|---|---|
| **Verification** | |
| General data | Author, title, date, research question |
| Theoretical framework | Adherence of the theoretical framework to the concept of responsible innovation |
| Context of the research | Country, sector, place of application |
| Method | Quality of the case study, unit of analysis, number of cases, sampling, techniques of data collection and analysis, data sources, validation |
| **Extraction** | |
| Analysis and interpretation of the data | Main contributions found, identification of elements or constructs, identification of frameworks |
| Discussion | Contributions to the advancement of the theory of responsible innovation |
| Conclusion | Theoretical contribution |
| **Overall article rating** | |
| General evaluation | Relevance to the study topic, quality and reliability of the study |

In general, the studies are focused on bioeconomy, like the biofuel sectors [75,76], sanitation [77], foods [78], and health. In the latter theme, the studies deal with diseases such as dementia [79], public safety through neuroimaging [80], and the process of developing new drugs [35]. All the studies are conducted in developed countries, such as Germany, Canada, Denmark, Scotland, the Netherlands, and the United Kingdom, and they achieve the proposed objectives and present results based on stakeholder inclusion. Only the study by Raman et al. [76] does not present the contribution of stakeholders clearly. Some studies also manage to generate results for specific stakeholders, such as Decker et al. [79], who, through workshops with several participants, present possibilities for product improvement for technology developers. The study by de Jong et al. [80] points to a change in perception of one stakeholder group over another group. Finally, the study by Bremer et al. [78] uses stakeholder inclusion to develop policies. The authors, however, highlight the need for the inclusion of these same stakeholders in the initial stages of the process, corroborating Burget et al.'s [3] and Owen et al.'s [7] proposal.

*3.5. Analysis at Individual Case Level*

To analyze the studies under synthesis, it was first necessary to analyze each case individually in order to achieve the meta-synthesis's main goal. To understand the role of stakeholders in the context of responsible innovation, as proposed by Hoon [70], a causal network was developed for each study. Based on the objective of the study and the four questions raised from the theory (described in Section 2.2), the following criteria were proposed: stakeholders involved, motivation for the inclusion of stakeholders, purpose of inclusion, agent of inclusion, innovation process phase, innovation form (4Ps), outcomes from stakeholders' inclusion, and highlights of the study. The results of this analysis are presented in Appendix B.

In all the studies, multiple stakeholders are involved, prioritizing the discussion and exchange of ideas between them and not just holding discussions individually within each group. Also included are stakeholders who would be considered unable to contribute, such as patients with dementia, even though the limitations resulting from this contribution are pointed out [79]. Further analysis is carried out in the next step.

### 3.6. Synthesizing Stakeholder Inclusion in the Analyzed Cases

Following the analysis at case level, we proceeded to sequencing the variables identified in each case and combined them into a single analysis. Figure 3 presents the findings following the 3W1H that we raised in the theoretical part, that is: (1) Who are the agents who orchestrate stakeholders' inclusion?; (2) When do stakeholders participate, at which stage of the innovation process does this happen?; (3) How does stakeholders' inclusion contribute to innovation—and which kind of innovation?; and (4) Who are the stakeholders who participate in the RRI process?

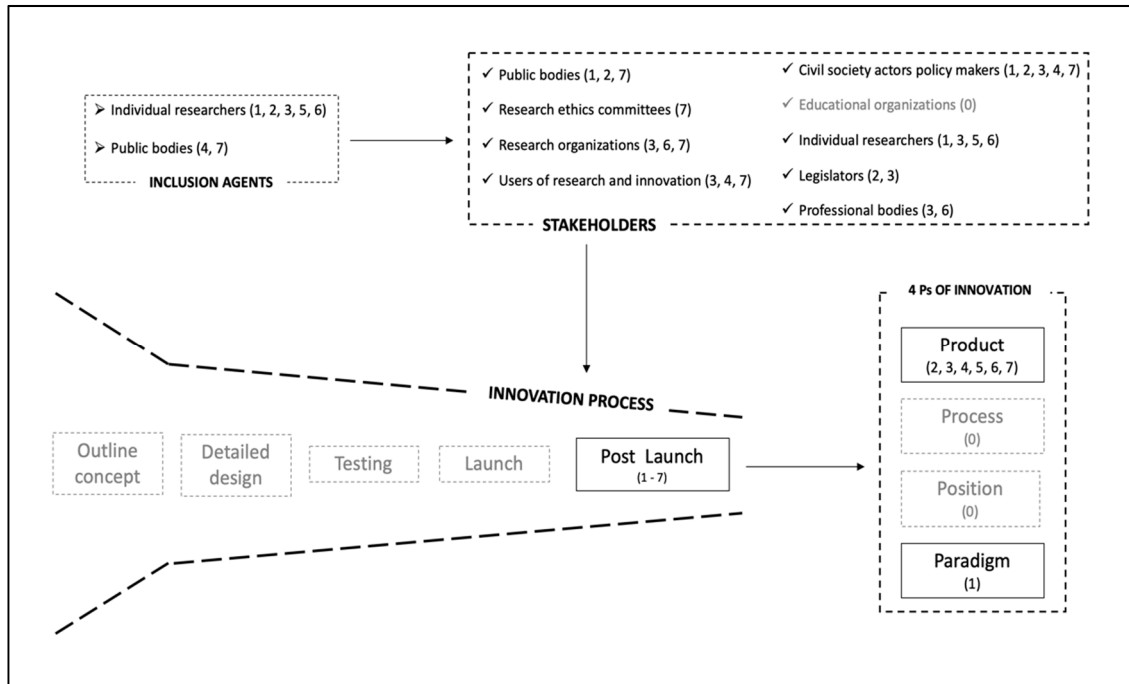

**Figure 3.** Stakeholder inclusion in the context of responsible innovation (number of the study in brackets).

Figure 3 demonstrates that agents like individual researchers or public bodies call for multiple stakeholders' inclusion, which occurs at the post-launch phase of the innovation process, contributing to new products or paradigm innovation.

Table 4 lists the studies referred to in Figure 3.

**Table 4.** Selected articles.

| Article | Title | Author | Year |
|---|---|---|---|
| 1 | Are plants the new oil? Responsible innovation, biorefining, and multipurpose agriculture | Shortall; Raman; Millar [75] | 2015 |
| 2 | Breaking barriers for a bio-based economy. Interactive reflection on monitoring water quality | Metze; Schuitmaker; Bitsch; Broerse [77] | 2017 |
| 3 | Integrating social and value dimensions into sustainability assessment of lignocellulosic biofuels | Raman; Mohr; Helliwell; Ribeiro; Shortall; Smith; Millar [76] | 2015 |
| 4 | Imagined technology futures in demand-oriented technology assessment | Decker; Weinberger; Krings; Hirsch [79] | 2017 |
| 5 | Inclusive deliberation and action in emerging RI practices: the case of neuroimaging in security management | De Jong; Kupper; Broerse [80] | 2016 |
| 6 | Responsible research and innovation: a productive model for the future of medical Innovation | Demers-Payette; Lehoux; Daudelin [35] | 2016 |
| 7 | Responsible techno-innovation in aquaculture: Employing ethical engagement to explore attitudes to GM salmon in Northern Europe | Bremer; Millar; Wright; Kaiser [78] | 2015 |

The fact that the selected papers refer to the research level corroborates that empirical evidence is still scarce [18] to the organizational level. The findings of the meta-synthesis are discussed in the next section.

## 4. Discussion

Performing the meta-synthesis enabled us to answer the questions raised through the literature review. In this section we will highlight our findings. We reinforce that we are proposing a connection between RRI and innovation management. Our selected cases are nested in academic environments and findings are specific to that particular context.

Our first question addresses the agents of inclusion: Who are the agents who orchestrate stakeholders' inclusion? In the selected cases, the agents who stimulate stakeholders' inclusion are academic researchers and researchers linked to multi-institutional projects. As demonstrated by Demers-Payette et al. [35], Shortall et al. [75], Raman et al. [76], Metze et al. [77] and de Jong et al. [80], academic researchers stimulate stakeholder inclusion. The motivation for such inclusion is the need to get better biofuels or health systems in the light of RRI. Bremer et al. [78] and Decker et al. [79] studies are part of large research studies developed through multi-institutional projects such as the Alliance Project and Pegasus.

Our findings constitute an important contribution towards RRI literature, since it is the first study that explicitly address and specifies the agents of inclusion. From the observation of the agents who motivated RRI in the analyzed studies, we obtain the first proposition:

**Proposition 1.** *The role of stakeholders in RRI nested in academic contexts is stimulated by academic researchers and members of institutional projects.*

At the organizational level, external agents will hardly influence the innovation process. On the other hand, when the agent is the innovator himself, or multi-institutional groups in which innovators are present, this can provide opportunities to enhance the RRI process. This is described by Decker et al. [79], who conclude that "involvement of technology developers helped these participants to begin imagining more specific potential technical solutions and to assess them with respect to their future desirability" (p. 177). In these cases, inclusion is linked to the dimension of reflexivity. The role of stakeholders is highlighted where building reflexivity of actors and institutions means rethinking the prevailing concepts relating to moral division of labor within science and innovation [81]. In this sense, mechanisms are explored to connect value systems external to science practices, such as codes of conduct, moratoria, and standards [13]. The findings of this study also highlight the contribution of stakeholders in the dimension of responsiveness, since this response consists of adjusting the course of action while acknowledging lack of knowledge and control [7]. It is important to emphasize that, in the context of responsible innovation in an organization (not research), the innovation process occurs within companies. This means that it is essential that innovators (e.g., R&D and marketing personnel) orchestrate the inclusion process. In fact, companies may consider stakeholders' knowledge as a resource that needs to be managed. It will be the combination of resources, capabilities, and managerial skills that makes it possible to achieve superior performance [62–64]. Through the orchestration, the decision-making process will be faster and more efficient while attending to reflexivity and responsiveness.

Despite the relevance of innovators as inclusion agents, this raises the question of how we can stimulate other agents to be part of the inclusion process. In the same way, how do agents orchestrate stakeholder inclusion?

Besides the indication that the inclusion should occur at the early stages of innovation [3], we raised the following question: When stakeholders participate, at which stage of the innovation process does this happen? From the results of the studies, it can be concluded that inclusion is carried out in the final stage of innovation, that is, when the product or service is already in the market.

Inclusion therefore serves as the basis for modifications in the developed product. Decker et al. [79] found that the involvement of technology developers helped participants to begin to imagine specific potential technical solutions and to evaluate them in relation to their future desirability.

After the meta-synthesis was finished, we identified three studies in which the stakeholders are involved in the first phase of research or innovation. However, in those studies, inclusion was not being seen as a strategic business process [18,82]. On the other hand, we identified research where people with mental disabilities and autism spectrum disorder participate, from the beginning of the project, in the development of games with real life application and virtual reality [83]. By including them at the preliminarily stage, the researchers reinforce that, in this way, it is possible to help address their specific needs and preferences. This will potentially deliver a range of broader social and individual benefits, such as improved literacy, socialization, and communication skills. We did not find any other studies on this subject and therefore we cannot affirm that inclusion at the initial stage occur in practice. However, we are aware of organizations where stakeholders are included at the initial stages of innovation and product development. The Norwegian company Laerdal Medical, which develops mannequins for training and simulation of resuscitation and birth, has a history of stakeholder participation throughout the innovation and development process [20]. This model avoids the loss of positive influence on technological development, which generates efforts to develop long-term thinking capacity, ethical reflection and efforts to shape the R&D course [84].

In the context of research, the second proposition is presented:

**Proposition 2.** *The role of stakeholders in academically nested RRI takes place when the innovation is already in the market.*

Whilst inclusion of stakeholders at market launch brings value, the potential benefits of such inclusion seem to be limited. Our findings indicate a lack of stakeholders' involvement in the early innovation stages, which ultimately leads to a lack of anticipation [7]. This results in being "locked into" a dominant design too early and does not allow enough flexibility to avoid losses when the need for adjustments becomes clear at a later stage of product or service development. However, keeping the design space open and flexible is not an easy task and is affected by institutional and cultural resistance [85].

There are several suggested methods for anticipation, such as foresight, technology assessment, horizon scanning, and scenario planning, as well as socio-literary techniques based on science fiction [7]. In all these methods, it is possible to perceive an advisory role of the stakeholders. Accordingly, we questioned how stakeholders can effectively co-create with innovators, participating in as many steps as possible from the beginning. Insights from innovation management literature here might provide useful guidelines, introducing innovation management techniques such as agile method, design thinking, project management, open innovation, and other methods [22,24,25,47,48]. This seems not to be reflected upon in RRI literatures when enhancing stakeholder participation in RRI.

Considering the four 'P's of innovation, which summarize the four dimensions of change—product, process, position and paradigm [26,66]—we raised another question: How does stakeholders' inclusion contribute to innovation—and which kind of innovation? Decker et al. [79] find that involvement of technology developers helps these participants to begin to imagine more specific potential technical solutions and to evaluate them in relation to their future desirability. A change in the paradigm of innovation is noticed by Shortall et al. [75]. In their study, consulting stakeholders resulted in the development of an alternative agriculture, predicting the sustainable production of multifunctional biomass in terms of a nutrient and energy cycle in the farm and local production on a smaller scale. However, it is concluded that the inclusion was merely advisory, with no evidence that the suggestions resulted in improvements in innovations. Thus, the evidence from the empirical studies analyzed suggests limited stakeholder contribution to product and paradigm innovations.

In this sense, innovation management tools, like design thinking, would help agents to better explore stakeholder knowledge.

We identified stakeholder contribution as being limited to product and paradigm innovation in the studies we analyzed. Although innovation can occur in products, processes, positions, and paradigms [26,66], the contribution of stakeholders so far identified from empirical cases does not influence the processes or the position of innovation. One explanation of this phenomena is that because the process and position are, intrinsic to the organization and can reveal strategic orientations they cannot easily be implemented, or are not in the company's interest, due to the risk of strategic exposure. This inference is supported by Block [18] and Balka [68], who discuss selective openness. This is whereby organizations protect internal knowledge to avoid a loss of control and a reduction of competitive advantage.

When considering where innovation takes place, the third proposition is found:

**Proposition 3.** *The role of stakeholders in RRI involves limited product and paradigm innovations.*

This proposition raises questions to be addressed in future research, such as how companies would include stakeholders to contribute to innovation in position, or even in process, and how stakeholders can contribute to that kind of innovation.

Finally, aiming to know the role of stakeholders in the RRI process, we raised the last question: Who are the stakeholders who participate in the RRI process? A wide variety of stakeholders were involved in the analyzed cases. These includes biomass producers, government representatives, biomass industry representatives, academics, NGO representatives [75]; representatives of the Ministry of Economic Affairs, nature and environment foundations, environmental education foundations, the OECD [77]; stakeholders in agriculture and intermediate work related to farms, bioenergy science, research and industry, policy makers and NGOs, and a group of experts (five leading members of the bioenergy research community with expertise in life sciences, life cycle assessment, sustainability assessment, and social sciences) [76]; patients (with dementia), relatives of patients, professional caregivers, volunteers [79]; scientists using neuroimaging technologies, security professionals, neuroscientists, social psychologists, security professionals, development practitioners, and an ethicist [80]; researchers (engineers and designers), innovation managers (universities, health organizations and biomedicine companies) [35]; and researchers, bioethics experts in aquaculture, industry representatives, seafood technology experts, lifetime patent attorneys, government representatives, animal preservation NGO representatives, veterinarians, and fish breeders [78]. Arguably, it is possible to develop a general empirical typology of the actors: producers, policy makers, researchers, scientists, intellectual patent managers, and direct and indirect users.

We established that RRI inclusion covers multiple stakeholders. The innovation agents encouraged the inclusion of policy makers (including funding agencies, regulators, and executives), business/industry representatives (internal or outsourced innovation departments and/or some R&D bases), civil society organizations (such as foundations, associations, social movements, community organizations, charities, media) as well as researchers and innovators (affiliates of various institutions and organizations at different levels).

The inclusion was aimed at exploring the ethical and social aspects [75,78] as well as stimulating the actors to imagine options for desirable technological futures [35,76,77,79,80]. What should be highlighted is the selection of stakeholders. In each study, external and internal stakeholders (in this case, those responsible for the development of technology), both economic and non-economic, are invited. The selection is made based on the potential contribution of each one, even though they may have divergent views [78,80]. Conflicts of interest [67], divergent views [18], and information asymmetry [85] are challenges inherent to the process but should be seen as an important step in creating a networked dialogue [86]. Inclusion generates new perspectives, such as new production

models [75,76,79], highlighting the need for anticipation [35], and contributing to the development of public policies [78].

Some studies do not provide details about the results of stakeholder inclusion [35,76,77]. Different techniques were used for inclusion, such as interviews [76,77,80], collective dialogue sessions [77,80], focus groups [35,80], workshops [78] and discussions with groups of experts [76,79]. In some cases, more than one method was used, which produced a more thorough result [76,79,80]. When the methods of inclusion involved groups, the contribution of the participants was also clearly the best [77,79,80]. In such cases, it was possible for the participants to interact regarding the proposed problem and to develop solutions together. As pointed out in the literature, the divergence of opinions as well as the overlapping of views were perceived by the facilitators who conducted the activities. It was also necessary for distinct groups to become familiar with each other, trying to find common 'ground' to start the discussion [80].

Despite the efforts, however, some negative points are highlighted. Some stakeholders felt they were not able to influence policies, guidelines and the general public [77]. Some brought up questions that were not under discussion as well as giving few or vague answers [79]. The facilitators themselves highlighted difficulty in ensuring quality conversations when there is a very diverse group of stakeholders, as it requires many elements to be dealt with simultaneously [80]. In addition, stakeholders do not always perceive value in their inclusion and consider the exercise to be without purpose. Consequently, the lack of motivation of a particular group ends up discouraging others [80]. Therefore, with regard to the stakeholders involved, the fourth and final proposition can be presented:

**Proposition 4.** *External stakeholders play mainly an advisory role in RRI, while internal stakeholders are truly involved in innovation process.*

While it is evident that multiple stakeholders are involved in RRI, their roles are different in relation to their contribution in the innovation process. Our observation points to the fact that although a broad range of external stakeholders were involved in the projects observed, their contribution was limited as they mostly had an advisory role. This calls to future studies on gaining understanding of how knowledge and viewpoints of external stakeholders can be better integrated into the innovation process.

## 5. Conclusions

The purpose of this article was to analyze the role of stakeholders in the context of responsible research and innovation. This is important in view of global grand challenges and Sustainable Development Goals since stakeholders are central to the achievement of responsible outcomes.

Our meta-synthesis revealed that stakeholder inclusion is debated in different yet related literature steams—RRI discourse as well as innovation management discourse. Whilst for both discourses the transfer of knowledge between different parties is crucial, innovation management literature emphases user and consumer involvement, while RRI literature is looking for broader stakeholder engagement. At the same time, we found that it is worth applying lenses of innovation management process towards studies of stakeholder's role in RRI. Thus, our main goal was to contribute to and to extend knowledge within the RRI domain by bridging it with different but related literature stream(s) and applying this new knowledge to the analysis of responsible research and innovation processes described in recent empirical RRI studies.

By adapting an innovation management process view to RRI perspective, this study contributes to RRI debate by highlighting the agents of stakeholder involvement, the stage of the stakeholder involvement in innovation process, types of innovation to which those stakeholders contribute to and types of stakeholders who participate or are eliminated from the innovation process. Our major theoretical contribution is in integrating an innovation process 'funnel' view to responsible governance

of research-based innovations and identifying the role of economic as well as non-economic stakeholders in the innovation process.

We established that the role of stakeholders in the research context of RRI is motivated mostly by academic researchers and members of institutional projects, classified as external and non-economic stakeholders. However, our findings reveal that stakeholders are often included when the innovation is already in the market, thereby curtailing stakeholders' role in the innovation process.

Most empirical studies of RRI are concerned with product and paradigm innovations, and the selected studies demonstrate the involvement of multiple stakeholders, apart from representatives of educational institutions. This is remarkable, given that educational institutions play a key role in educating future innovation users. Within the context of the research, the studies all stem from developed countries, such as Germany, Canada, Denmark, Scotland, the Netherlands, and the United Kingdom.

Our study also has implications for practitioners. Although our findings are limited to responsible innovations that emerged in academic settings, we identified several challenges in stakeholder involvement in such projects. One major challenge is late inclusion that limits the anticipation of risks as well as reflection of stakeholders' feedback. Late inclusion considerably increases risk of innovation rejection and increases the cost of adoption of innovation once in the market. Thus, a practical application following from our findings is for governance of research-based innovations, that often emerge in an academic context. There is a need to develop a better understanding of the importance of early stakeholder involvement and its implications for the whole innovation process.

Furthermore, our study identified several challenges in the inclusion process, such as power imbalance, conflict of interest and difficulty of orchestrating inclusion to achieve acceleration of innovation process and responsible outcomes. This demonstrates that agents of involvement in innovation should familiarize themselves with available tools and techniques that are successfully applied in innovation management at the organizational levels. However, one should bear in mind that these tools might not be developed to deliberate inclusion and should be carefully adapted to become instrumental.

*Limitations and Avenues for Future Research*

As any study, our research is not without limitations. First, we constrained our findings to peer-review literature. While such approach ensures the high quality of analyzed articles, it also limits us in quantity of empirical findings. Certainly, employing different type of literature review, for example scoping review [86] which includes 'grey' literature would allow to increase the number of empirical findings, and can be a valuable exercise for future research.

In our search we applied certain key words, which at the stage of the study design seemed most appropriate for the purpose of the research. However, a different combination of words might yield more and different kinds of literature. Our decision was guided by the intention to contribute to RRI literature, while for example not covering terms on user inclusion that are widely discussed in innovation management due to differences in vocabulary in these two distinct scientific fields. As most RRI studies focus on the academic or policy environment [16,18], our sample of studies are conducted within the academic environment and based on projects developed from within academia. Thus, the findings of this review are limited to this particular environment and cannot be generalized to all innovation processes aiming for responsible outcomes within a diverse set of organizations. Any future research should attempt to cover issues relating to stakeholder inclusion across different empirical and theoretical studies, including RRI, innovation management, CSR, and STS studies. This would allow for generalizable evidence on the role of stakeholders in the innovation process for the purpose of responsible outcomes. We also suggest the analysis of the inclusion on the development of new business models, societal innovations, among others, once is not only product/service development process that would be positively impacted by the stakeholder inclusion. RRI literature reinforces a dichotomy between secrecy and transparency, which generates a selective opening of information.

This is due to the fear of reducing competitive advantage, loss of control over the product and fear of knowledge leakage. Further research could investigate the consequences of transparency versus patents level, mainly at the first steps of the innovation process, regardless the patent costs.

Based on our findings, we also point to some unclear aspects that can inspire further research. We highlight that the literature just presents the manager as the agent and does not explore abilities or capabilities [87], even when discussing leadership [88]. The orchestration of stakeholders and other resources can shed light on the way in which these agents work. In the same way, future research should address the reasons for late inclusion at the research stage and the impact on innovation. Future research can also identify potential mechanisms for the early involvement of diverse stakeholders in the innovation process. The degree of stakeholders' contribution can also be researched, as it was not considered in this study; also, the potential for a stakeholder creating nuisance will greatly influence the way the stakeholder is included. We also consider that the psychology of stakeholders is an important issue for further research, to what degree will their behavior, pre-conceptions, and values influence their contribution. Finally, it is not always the case that single organizations/projects should be responsible for and can actually exercise all the principles of RRI. The eco-system on the regional, national or even international level should be supportive of stakeholder inclusion, among other things, to ease this process for single actors. In this sense, we suggest conducting research on policies that may contribute to the inclusion process. Lastly, we suggest studies in the context of developing countries since it is shown to be a research gap at the same time as playing a fundamental role in the SDGs.

**Supplementary Materials:** The supplementary materials are available on https://www.researchgate.net/publication/331980818_The_Role_of_Stakeholders_in_the_Context_of_Responsible_Innovation_A_Meta-Synthesis at Research Gate.

**Author Contributions:** L.M.d.S. was the leading author of this review. C.C.B. contributed significantly to the study design and the synthesis of the results. K.F. contributed significantly to the synthesis of the results and the final writing of the review. T.I. contributed significantly to the conceptual analysis of the framework and the final writing of the review.

**Funding:** This research received no external funding.

**Acknowledgments:** The authors are grateful for the financial support for this research provided by the Brazilian agency CAPES Foundation of the Ministry of Education.

**Conflicts of Interest:** The authors declare no conflicts of interest.

## Appendix A. List of Publications

| Total | Publication |
|---|---|
| 78 | *Journal of Responsible Innovation* |
| 8 | *Science & Public Policy (SPP)* |
| 4 | *Science and Engineering Ethics* |
| 3 | *Asian Biotechnology & Development Review* |
| 3 | *Design Issues* |
| 3 | *Science & Engineering Ethics* |
| 3 | *Science and Public Policy* |
| 2 | *Bized* |
| 2 | *Cyberpsychology, Behavior & Social Networking* |
| 2 | *IEEE Technology and Society Magazine* |
| 2 | *Information Polity: The International Journal of Government & Democracy in the Information Age* |
| 2 | *International Conference on Research Challenges in Information Science (RCIS)* |
| 2 | *Journal of Business Ethics* |
| 2 | *Journal of Nanoparticle Research* |
| 2 | *Nanoethics* |
| 2 | *Research Policy* |
| 2 | *Social Sciences* |
| 2 | *Technological Forecasting and Social Change* |

| Total | Publication |
| --- | --- |
| 1 | *Academic Pediatrics* |
| 1 | *Academy of Management* |
| 1 | *Annals of the Association of American Geographers* |
| 1 | *Aquaculture* |
| 1 | *Biofabrication* |
| 1 | *Biomass & Bioenergy* |
| 1 | *Biometric Technology Today* |
| 1 | *BMC Family Practice* |
| 1 | *BMC Health Services Research* |
| 1 | *Business and Non-Profit Organizations Facing Increased Competition and Growing Customers' Demands, Vol. 11* |
| 1 | *Chemistry Education Research and Practice* |
| 1 | *Competition Forum* |
| 1 | *Concepts & Transformation* |
| 1 | *Conference on Intellectual Capital/Academic Conferences & Publishing International* |
| 1 | *Current Opinion in Obstetrics and Gynecology* |
| 1 | *Daily Compilation of Presidential Documents* |
| 1 | *Economic Review* |
| 1 | *Energy Policy* |
| 1 | *Environmental Health Perspectives* |
| 1 | *Environmental Science & Policy* |
| 1 | *Equitybites (M2)* |
| 1 | *European Education* |
| 1 | *European Journal of Human Genetics* |
| 1 | *European Planning Studies* |
| 1 | *Evidence & Policy: A Journal of Research, Debate & Practice* |
| 1 | *Evidence Based Midwifery* |
| 1 | *Global Banking News (GBN)* |
| 1 | *Greener Management International* |
| 1 | *Health Research Policy and Systems* |
| 1 | *Higher Education Management & Policy* |
| 1 | *Higher School's Pulse* |
| 1 | *Human Geographies—Journal of Studies & Research in Human Geography* |
| 1 | *Information Society* |
| 1 | *Innovation: The European Journal of Social Sciences* |
| 1 | *International Conference on Applied Human Factors and Ergonomics (AHFE)* |
| 1 | *International Conference on Education and Educational Conference* |
| 1 | *International Conference on Engineering, Project, and Production Management* |
| 1 | *International Forum on Knowledge Asset Dynamics: Knowledge and Management Models for Sustainable Growth* |
| 1 | *International ICE Conference on Engineering, Technology and Innovation (ICE)* |
| 1 | *International Journal of Action Research* |
| 1 | *International Journal of Critical Infrastructures* |
| 1 | *International Journal of Performability Engineering* |
| 1 | *International Journal of Science in Society* |
| 1 | *International Journal of Technology Assessment in Health Care* |
| 1 | *International Journal of Technology Management* |
| 1 | *International Journal of Voluntary & Nonprofit Organizations* |
| 1 | *International Management Conference: New Management for the New Economy* |
| 1 | *Internet of Things: IoT Infrastructures, Pt I* |
| 1 | *Jcom—Journal of Science Communication* |
| 1 | *Journal of Agricultural & Environmental Ethics* |

| Total | Publication |
|-------|-------------|
| 1 | *Journal of Artificial Societies & Social Simulation* |
| 1 | *Journal of Baltic Science Education* |
| 1 | *Journal of Change Management* |
| 1 | *Journal of Construction Engineering & Management* |
| 1 | *Journal of Contingencies & Crisis Management* |
| 1 | *Journal of Decision Systems* |
| 1 | *Journal of Educational & Psychological Consultation* |
| 1 | *Journal of Evaluation in Clinical Practice* |
| 1 | *Journal of Innovation Economics & Management* |
| 1 | *Journal of Law and the Biosciences* |
| 1 | *Journal of Law, Medicine & Ethics* |
| 1 | *Library Leadership & Management* |
| 1 | *Military Medicine* |
| 1 | *Minerva: A Review of Science, Learning & Policy* |
| 1 | *Multidimensional Education & Professional Development. Ethical Values* |
| 1 | *New Approaches in Social and Humanistic Sciences* |
| 1 | *Nutrition Bulletin* |
| 1 | *Ocean & Coastal Management* |
| 1 | *Omics—A Journal of Integrative Biology* |
| 1 | https://home.liebertpub.com/publications/cyberpsychology-behavior-brand-social-networking/10/overview |
| 1 | *Organisational Learning in the Automotive Sector* |
| 1 | *Physiological Society* |
| 1 | *Plos Biology* |
| 1 | *Policy Research* |
| 1 | *Policy Studies Journal* |
| 1 | *PR Newswire* |
| 1 | *Prometheus* |
| 1 | *Public Roads* |
| 1 | *RAI—Revista de Administração e Inovação* |
| 1 | *Reference & Research Book News* |
| 1 | *Research Evaluation* |
| 1 | *Risk Analysis* |
| 1 | *Robotics and Autonomous Systems* |
| 1 | *Scandinavian Journal of Forest Research* |
| 1 | *Socialiniai Tyrimai* |
| 1 | *Society and Business Review* |
| 1 | *Sociology Compass* |
| 1 | *Stanford Social Innovation Review* |
| 1 | *Studies in Higher Education* |
| 1 | *Substance Use & Misuse* |
| 1 | *World Conference on Technology, Innovation and Entrepreneurship* |
| 1 | *World Future Review (World Future Society)* |
| 1 | *Wroclaw University of Economics* |

## Appendix B. Coding Form

| Article | 1 | 2 | 3 | 4 |
|---|---|---|---|---|
| **Title** | Are plants the new oil? Responsible innovation, biorefining and multipurpose agriculture | Breaking barriers for a bio-based economy: Interactive reflection on monitoring water quality | Integrating social and value dimensions into sustainability assessment of lignocellulosic biofuels | Imagined technology futures in demand-oriented technology assessment |
| **Authors** | Shortall; Raman; Millar | Metze; Schuitmaker; Bitsch; Broerse | Raman; Mohr; Helliwell; Ribeiro; Shortall; Smith; Millar | Decker; Weinberger; Krings; Hirsch |
| **Published** | *Energy Policy* | *Environmental Science & Policy* | *Biomass & Bioenergy* | *Journal of Responsible Innovation* |
| **Year** | 2015 | 2017 | 2015 | 2017 |
| **Stakeholders** | Biomass producer, government representative, biomass industry representative, academic, NGO representative | Representative of the Ministry of Economic Affairs, foundation of nature and environment, environmental education foundation, OECD, foundation of quality infrastructure and soil management control, water cycle research institutes, researchers, water and soda producer | Parties interested in agriculture and intermediate work related to farming, bioenergy science, research and industry, policy makers, NGOs, and expert groups | Patients (with dementia), patient relatives, caregivers, volunteers, technology developers |
| **Motivation for the inclusion** | The study is part of a larger project that explores the ethical and social issues raised by the use of perennial energy crops and crop residues in energy production in the UK and Denmark. | (a) Engage stakeholders in a conversation about the past, present and future—that is, contextualize niche innovations; and (b) involve them in deliberations on the barriers coming from that context and internally | Develop future visions for technology | Develop a constructive dialogue between technological innovators, users and other stakeholders |
| **Purpose of inclusion** | Explore the ethical and social aspects of energy production | Participants interactively reflect these barriers and collaboratively develop alternatives to deal with them. | Cross-referencing information with other data sources | Encourage actors to imagine options productively for desirable technological futures |
| **Agent of inclusion** | Academic researchers | Academic researchers | Academic researchers | Multi-institutional project |
| **Innovation process phase** | Post-launch | Post-launch | Post-launch | Post-launch |
| **Innovation form (4 Ps)** | Paradigm | Product | Product | Product |
| **Outcomes from stakeholder inclusion** | Cross-utilization of industrial biorefins and alternative agricultural frameworks of multi-purpose biomass production systems, which is different from the clear cut-off between industrial and alternative views of the bioeconomy | No practical perspectives were presented on the subject. | Transformation of biomass into biorefineries; transformation of the agricultural system; evolution of agricultural practices. | From the stakeholder workshops, it was possible to develop solutions that could serve as the basis for a later technology development project. |
| **Highlights** | All the interviewees were focused on the product and not on the biorefining process; proposed to return to an old model of planting. | The results demonstrate that systemic barriers are discursively pronounced in niches—or other forms of responsible research and innovation—and may hinder change, even at the niche level; the problems pointed out are external to the organization into which the participants are inserted. | | Some of the solutions refer to new uses for existing technologies and improvements in widely used equipment. The main highlight, however, is the demonstration of the benefits of stakeholder involvement in the process of improving technological innovations. |

| Article | 5 | 6 | 7 |
|---|---|---|---|
| Title | Inclusive deliberation and action in emerging RI practices: The case of neuroimaging in security management | Responsible research and innovation: A productive model for the future of medical innovation | Responsible techno-innovation in aquaculture: Employing ethical engagement to explore attitudes to GM salmon in Northern Europe |
| Authors | De Jong; Kupper; Broerse | Demers-Payette; Lehoux; Daudelin | Bremer; Millar; Wright; Kaiser |
| Published | *Journal of Responsible Innovation* | *Journal of Responsible Innovation* | *Aquaculture* |
| Year | 2016 | 2016 | 2015 |
| Stakeholders | Scientists using neuroimaging technologies, security professionals, neuroscientists, social psychologists, security professionals, development professionals and an ethicist | Users of medical technologies (patients and clinicians), developers (engineers and designers), innovation managers (universities, health organizations and biomedicine companies) | Researchers, bioethics, aquaculture expert, industry representative, seafood technology expert, lifetime patent attorney, government representative, animal preservation NGO representative, veterinarians, fish breeder |
| Motivation for the inclusion | Creating scenarios about the future of neuroimaging | Develop an approximation between the concept of IR and health systems | Part of the European Commission's research on the public perception of genetically modified animals, which aims to serve as a basis for the development of future EU policies |
| Purpose of inclusion | Develop the interaction of different stakeholders | Think productively about R&D in the medical field | Map and analyze the ethical aspects of GM animals by European stakeholders |
| Agent of inclusion | Academic researchers | Academic researchers | Multi-institutional project |
| Innovation process phase | Post-launch | Post-launch | Post-launch |
| Innovation form (4 Ps) | Product | Product | Product |
| Outcomes from stakeholder inclusion | The resulting imagery needs broader molding effects by those who do not necessarily need to gain from the creation of technology. Stakeholders, such as public authorities, non-governmental organizations and the general public, should also be included. | It proposes the implementation of a responsible innovation approach to health care but requires developers to anticipate carefully the consequences and opportunities associated with medical innovations, including their organizational, social, economic, and ethical issues. | Participants at all three workshops in Germany, Norway, and the United Kingdom encouraged more participatory decision-making processes involving all aquaculture stakeholders to decide the future of new technologies, agreeing that all forms of evidence should be heard, ranging from scientific risk assessments to traditional knowledge systems, religious concerns or even stakeholder values. |
| Highlights | Study for scenario construction on the future of neuroimaging | They bring the following questions: Can the application of the four dimensions bridge the gap between these worlds and lead to the creation of a lasting shared world for responsible innovation? Which theoretical lenses should be mobilized to make sense of these shared and non-linear practices | Project for the establishment of public policies |

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
