# Peer review of "The Role of Stakeholders in the Context of Responsible Innovation: A Meta-Synthesis"

_sustainability, doi:10.3390/su11061766_

Round 1

Reviewer 1 Report

On the positive: I find the topic relevant, the method sound, and the conclusions interesting. 

On the negative: I found the paper rather longwinding relative to what it had to say: a literature review of Stakeholder involvement in RI. The abstract says it all.

The paper reads somewhat repetitive (there are five or so instances of summing-up the who-when-how-who... that feels like several too many), possibly because it documents the entire method of meta-review (217 papers > 61 papers > 55 papers > 7 papers).

The result feels a bit like an anti-climax; 7 papers, with relatively little in common content-wise (biofuel, sanitation, foods, dementia, neuroimaging, drugs development)--which raises questions about generalizability. So we found (only!) 7 papers, 3 of which from the usual suspect, i.e. the Journal of Responsible Innovation. Was that worth the extensive meta-review process? 

One limitation of the literature review is (as the authors point out, line 466) is that they only looked at published articles in scientific journals. Maybe they could have looked, in only briefly, at conference proceedings (e.g. Participatory Design ... concerns similar to Responsible Innovation), (popular) books (e.g. Open Innovation ... about involving stakeholders) or brochures from corporations who champion Responsible Innovation or stakeholder participation, or design or innovation agencies. There must be more information out there. 

I feel the paper could benefit from a partial re-write: to make it less repetitive/more compact; and to include some other/non-scientific literature (not a new meta-analysis, but some examples). 

Minor: there seem to be words missing in line 517, just before "(such as foundation ..."

Author Response

Dear Reviewer,

We thank you and the other reviewers for the comments and contributions. You have allowed us to considerably improve our paper. We have now uploaded all the documents again via the submission system.

We have endeavoured to attend all the reviewers' recommendations, which have proven to be very useful and coherent. The responses to each of your recommendation or suggestion can be found below.

Before responding to all reviewers’ comments individually, we would like to explain the main changes we made in the paper. These key modifications were motivated by observations that were repeated, in different forms, by all reviewers:

We reorganized the abstract, better      detailing the goal and results of the paper, as our contribution to RRI      literature;

We reorganized the introduction,      seeking to make the paper's contribution clearer;

We insert suggestions of the      evaluators in the literature review, seeking a better positioning for our      contribution;

4.     We reorganized the sub-section Synthesizing stakeholder inclusion in the analysed cases and the discussion of the results, aiming to be more objective;

We reorganized      conclusion, also seeking to make the paper's contribution clearer and      instigating for further research.

Once again, thank you for your fruitful comments, it helped a lot to improve our paper

Best regards,

Authors

Point 1: I found found the paper rather long winding relative to what it had to say: a literature review of Stakeholder involvement in RI. The abstract says it all.

Response 1: Thank you for calling our attention to this. We made some improvements, especially on the sub-section Synthesizing stakeholder inclusion in the analysed cases and the discussion of the results, aiming to be more objective and reduce the size.

Point 2: The paper reads somewhat repetitive (there are five or so instances of summing-up the who-when-how-who... that feels like several too many), possibly because it documents the entire method of meta-review (217 papers > 61 papers > 55 papers > 7 papers).

Response 2: Thank you for bringing this to our attention. We reviewed the article in order to make it more objective. We removed replication where we noticed it and hope that it reads better now.

Point 3: The result feels a bit like an anti-climax; 7 papers, with relatively little in common content-wise (biofuel, sanitation, foods, dementia, neuroimaging, drugs development)--which raises questions about generalizability. So we found (only!) 7 papers, 3 of which from the usual suspect, i.e. the Journal of Responsible Innovation. Was that worth the extensive meta-review process?

Response 3: Thank you for your comments. We share the concern with the low number of articles. In this article, we applied the meta-synthesis (the analysis of the analysis). This method aims at building theory out of primary qualitative case studies that have not been planned, in other words, the use of the meta-synthesis allows the researcher to seek answers to his questions in other qualitative scientific studies, based on the interpretation of qualitative evidence from a post-positivistic perspective (Hoon, 2013). We think that a low number of empirical studies is due to the fact that RRI literature so far was quite normative, with a relatively low number of empirical studies per se. Another observation brought by reviewer 3 to our attention, is that the key search words we applied allowed us to find literature from RRI perspective only, while other domains (for example innovation management literature) use other terms while addressing the related issues. We have reflected on this concerns in the conclusion section of our article, in particular in section “limitations and future research”.

Point 4: One limitation of the literature review is (as the authors point out, line 466) is that they only looked at published articles in scientific journals. Maybe they could have looked, in only briefly, at conference proceedings (e.g. Participatory Design ... concerns similar to Responsible Innovation), (popular) books (e.g. Open Innovation ... about involving stakeholders) or brochures from corporations who champion Responsible Innovation or stakeholder participation, or design or innovation agencies. There must be more information out there.

Response 4: Thank you for your observation. In fact, the databases search revealed some conference proceedings like demonstrated in Table 1. However, why we recognize that other types of literature review (for example scoping review) would allow too much broader inclusion of “grey” literature, we stick to our chosen aim of including a verified research results that were subject to peer-review process and chosen method of analysis.

Point 5: I feel the paper could benefit from a partial re-write: to make it less repetitive/more compact; and to include some other/non-scientific literature (not a new meta-analysis, but some examples).

Response 5: Thank you for bringing this to our attention. We made a series of improvements to make it less repetitive. We also included some other examples specifically in the discussion and conclusion sections.

Point 6: Minor: there seem to be words missing in line 517, just before "(such as foundation ...”

Response 6: Thank you for calling our attention to this flaw. We included the missing term “civil society organizations”.

Reviewer 2 Report

General comments:

·        First, I am missing a theoretical/conceptual basis for the paper. This relates to my comment #24 further below. You talk, for example, about stakeholders but do not mention the Corporate Social Responsibility literature. Consequently, the gap in research isn’t obvious.

·        Second, I do not understand why you include the SDGs in your paper. There is no need for them; you could simply make the paper about stakeholder engagement in RI case studies. Also, they only come up in the title and the beginning of the paper but then only in one sentence of the discussion (line 554) and in the conclusion only by referring to literature again; i.e. not in connection to your findings.

Comments on specific parts of the paper:

1.      Abstract: From the third sentence on, the abstract is concise and informative. The first two sentences, however, do not fit the line of argumentation and feel artificially added in order to justify the inclusion of the SDGs in the paper title. If you want to make a connection to the SDGs, you could start with them (i.e. the broad picture) instead of stakeholder participation and then make the link back to first RI and only then to stakeholder participation/inclusion. (By the way, you use both terms – apparently as synonyms – in the abstract. Try and stick to one or make a clear distinction.)

2.      Keywords: I suggest to add “responsible innovation” and “stakeholder inclusion”.

3.      Lind 65: von Schomberg’s definition isn’t cited correctly; it should be “mutually” responsive.

4.      Introduction: The way the SDGs and RI are described in the introduction makes it sound as if there was a clear connection between the two. This is, at least to my understanding, not (yet) the case. To me, this connection needs to be argued by the authors.

5.      Introduction: You start out with RRI and then switch to RI without explanation. Include your line of reasoning for dropping the “research and” part.

6.      Introduction: Be clear about the (lack of?) differences between stakeholder inclusion, stakeholder participation and deliberation. You may also want to make a connection to the EC’s key “Public Engagement”. For the different terms used, you might find inspiration in Marschalek (2017) at https://www.zsi.at/object/publication/4498/attach/Marschalek_Public_Engagement_in_RRI.pdf or some other source.

7.      Lines 84-90: Move to beginning of section 3.

8.      Section 2: I would strongly suggest changing the title – and focus – of this section to something like “Stakeholders in responsible innovation”. In its current form, this section creates the impression that “inclusion” was at the center of the interest of this study. However, you later describe the search terms used to select studies, which do neither include “inclusion” nor “engagement”, “participation”, etc. To remain in line with your empirical work, your conceptual framework should revolve around stakeholders. You have them in the title, so that is fine.

9.      Lines 110-111: “The main goal of inclusion is to diminish the authority of experts, […]” à Is it? I would argue against this, and talk instead about democratic processes, diversity, creativity and giving a voice to potentially affected and potentially marginalized groups. Add explanation and references.

10.   Lines 119-136: You mention a range of points and then pick two (sub-sections 2.1 and 2.2) for further investigation. It is currently not clear why you chose these two and not others. You may further want to make the different aspects more explicit by e.g. using bullet points and then connecting them to the following sub-sections.

11.   Line 136: Technically, the “next section” would be section 3. You refer to the following sub-sections.

12.   Line 140: You list customers and suppliers as examples for internal stakeholders. To me, these would clearly be company-external stakeholders. Adapt or add a strong reference.

13.   Line 157: The title is not self-explanatory and only referred to – for readers to guess at its meaning – in line 200. I suggest to replace it with a title that is easily understandable and captures the content of the sub-section.

14.   Lines 177-183: You should refer to an innovation process model that describes different phases or stages of an innovation process; and then refer back to it when you describe your findings in section 3.6.

15.   Line 202: I’d suggest to change the title of section 3 to “Method”, “Approach” or similar and then move lines 84-90 to the beginning of this section.

16.   Line 203: Describe the goal of the study here.

17.   Line 240: To be exact, you should refer to the research on RRI and RI; based on the search terms you described earlier.

18.   Line 241: While you searched for articles about “stakeholders”, you use the results to draw a conclusion about “inclusion”. This, in my opinion, is not permissible. I strongly recommend to stick to drawing conclusions about stakeholders; see also my comment #8.

19.   Line 253: Table 1 is a list and doesn’t really present any analysis. How about starting the sentence as “Table 1 presents…”

20.   Section 3.6: Make sure to word findings in a way that does not suggest that you would generalize your findings. As an example, the sentence in line 367 could start like this: “In the seven case studies that were analysed, the agents who…” or “In the identified sample of case studies…”. Another example; in lines 500-501 you state that “…the contribution of stakeholders does not influence the processes and the position of innovation”; which you cannot certainly not state as a general truth based on your analysis of seven case studies.

21.   Lines 387-398 offer no answer to the question at which stage of the innovation process stakeholders are included! If you have no information about this, I would suggest removing the question itself from the study. See also my comment #14.

22.   Lines 399ff: Here you talk about “the RI literature”. Make sure to stick with the analysis of your seven sample cases in this section.

23.   Line 438: This might be a language issue, but in my perspective you can draw conclusions from evidence but you cannot not develop, or state, “evidence” based on a literature survey. They could be hypotheses to be further explored in future research.

24.   Section 4: In this section – as well as in the whole paper – you use terms such as “stakeholder”, “agent” and “inclusion” that carry very specific meaning in different academic communitites (such as e.g. the term stakeholder in the CSR literature). You should state clearly if you refer to any specific definition of these terms.

25.   Conclusion section: Again, make sure to only draw conclusions that are within the limitations of the study method; and phrase them accordingly. Refrain e.g. from talking about “most empirical studies” (line 577).

26.   Line 587-588: I do not see how this study is connected to the SDGs. I especially don’t see any contribution to their achievement.

Author Response

Dear Reviewer,

We thank you and the other reviewers for the comments and contributions. You have allowed us to considerably improve our paper. We have now uploaded all the documents again via the submission system.

We have endeavoured to attend all the reviewers' recommendations, which have proven to be very useful and coherent. The responses to each of your recommendation or suggestion can be found below.

Before responding to all reviewers’ comments individually, we would like to explain the main changes we made in the paper. These key modifications were motivated by observations that were repeated, in different forms, by all reviewers:

We reorganized the abstract, better      detailing the goal and results of the paper, as our contribution to RRI      literature;

We reorganized the introduction,      seeking to make the paper's contribution clearer;

We insert suggestions of the      evaluators in the literature review, seeking a better positioning for our contribution;

4.     We reorganized the sub-section Synthesizing stakeholder inclusion in the analysed cases and the discussion of the results, aiming to be more objective;

We reorganized      conclusion, also seeking to make the paper's contribution clearer and      instigating for further research.

Once again, thank you for your fruitful comments, it helped a lot to improve our paper

Best regards,

Authors

General comments:

Point 1: First, I am missing a theoretical/conceptual basis for the paper. This relates to my comment #24 further below. You talk, for example, about stakeholders but do not mention the Corporate Social Responsibility literature. Consequently, the gap in research isn’t obvious.

Response 1: Thank you for your comments. We made a series of improvements in the introduction, including a discussion about CSR and its relation to RRI. We also provide more information about the research gap.

Point 2: Second, I do not understand why you include the SDGs in your paper. There is no need for them; you could simply make the paper about stakeholder engagement in RI case studies. Also, they only come up in the title and the beginning of the paper but then only in one sentence of the discussion (line 554) and in the conclusion only by referring to literature again; i.e. not in connection to your findings.

Response 2: Thank you for your comments. We agree that the major focus of this paper is RRI. Hence, we decided to remove SDG from the title of the paper. However, we feel that RRI can be seen as a tool to address, so it helps society and innovators see “how” to target SDG. We tried to better describe this connection in our paper. In that way we highlighted the importance of stakeholder’s inclusion in order to promote the anticipation of societal needs, even in the organizational level.

Comments on specific parts of the paper:

Point 1:      Abstract: From the third sentence on, the abstract is concise and informative. The first two sentences, however, do not fit the line of argumentation and feel artificially added in order to justify the inclusion of the SDGs in the paper title. If you want to make a connection to the SDGs, you could start with them (i.e. the broad picture) instead of stakeholder participation and then make the link back to first RI and only then to stakeholder participation/inclusion. (By the way, you use both terms – apparently as synonyms – in the abstract. Try and stick to one or make a clear distinction.)

Response 1: Thank you for bringing this to our attention. We rewrite the abstract to make clearer the SDG connection with RRI. Regardless to participation/inclusion terms, we made the adjustments and used only “inclusion”.

Point 2:      Keywords: I suggest to add “responsible innovation” and “stakeholder inclusion”.

Response 2: Thank you for your suggestion. We included the terms.

Point 3: Lind 65: von Schomberg’s definition isn’t cited correctly; it should be “mutually” responsive.

Response 3: Thank you for your observation, but the term used by von Schomberg’s in that specific citation (p. 1) is “mutual”. When he presents the concept to RRI (p. 9), he then used the term “mutually”.

Point 4: Introduction: The way the SDGs and RI are described in the introduction makes it sound as if there was a clear connection between the two. This is, at least to my understanding, not (yet) the case. To me, this connection needs to be argued by the authors.

Response 4: As we answered on General comments Response 2, we made an effort to better connect SDG and RRI.

Point 5: Introduction: You start out with RRI and then switch to RI without explanation. Include your line of reasoning for dropping the “research and” part.

Response 5: Thank you for bringing this to our attention. We included an argument about the difference between RRI and RI. And once the meta-synthesis explored cases based on research, we decided to use term RRI throughout the paper, although we point that this broader term also includes RI.

Point 6: Introduction: Be clear about the (lack of?) differences between stakeholder inclusion, stakeholder participation and deliberation. You may also want to make a connection to the EC’s key “Public Engagement”. For the different terms used, you might find inspiration in Marschalek (2017) at:

https://www.zsi.at/object/publication/4498/attach/Marschalek_Public_Engagement_in_RRI.pdf or some other source.

Response 6: Thank you for your observation. We defined the use of the term “inclusion”, excluding the term “participation”. In the Introduction we also present the main characteristic of inclusion (that focuses on who to involve, during which stage of the innovation process, and whether the stakeholder network is representative). Once some seminal articles also use the term “deliberation”, we mentioned that its focuses on the decision-making process.  But for our study, the target was specifically “inclusion”. We also thank you for the book suggestion. We used it to improve the theory.

Point 7: Lines 84-90: Move to beginning of section 3.

Response 7: Thank you for your suggestion. We excluded these lines.

Point 8: Section 2: I would strongly suggest changing the title – and focus – of this section to something like “Stakeholders in responsible innovation”. In its current form, this section creates the impression that “inclusion” was at the center of the interest of this study. However, you later describe the search terms used to select studies, which do neither include “inclusion” nor “engagement”, “participation”, etc. To remain in line with your empirical work, your conceptual framework should revolve around stakeholders. You have them in the title, so that is fine.

Response 8: Thank you for your suggestion. Once the name of the first sub-section in this section already is “The stakeholder in RI”, we renamed the section with a broader title – “Responsible Innovation”.

Point 9: Lines 110-111: “The main goal of inclusion is to diminish the authority of experts, […]” à Is it? I would argue against this, and talk instead about democratic processes, diversity, creativity and giving a voice to potentially affected and potentially marginalized groups. Add explanation and references.

Response 9: Thank you for bringing this to our attention. We consider that is an important point, that is related, for example, to users/patient inclusion.  We included an extra sentence arguing it.

Point 10: Lines 119-136: You mention a range of points and then pick two (sub-sections 2.1 and 2.2) for further investigation. It is currently not clear why you chose these two and not others. You may further want to make the different aspects more explicit by e.g. using bullet points and then connecting them to the following sub-sections.

Response 10: Thank you for the observation. In sub-section 2.1 we mentioned about the critical aspect about stakeholders: (1) the definition of who need to (and can) be included and (2) how they can really contribute to innovation. In fact, in the next sub-section (2.2) we develop both aspects. We rewrite in order to clearly each one.

Point 11: Line 136: Technically, the “next section” would be section 3. You refer to the following sub-sections.

Response 11: Thank you for calling our attention to this flaw. We adjusted it.

Point 12: Line 140: You list customers and suppliers as examples for internal stakeholders. To me, these would clearly be company-external stakeholders. Adapt or add a strong reference.

Response 12: Thank you for bringing this to our attention. We rewrite the phrase.

Point 13:   Line 157: The title is not self-explanatory and only referred to – for readers to guess at its meaning – in line 200. I suggest to replace it with a title that is easily understandable and captures the content of the sub-section.

Response 13: Thank you for your observation. We kept the title, but complementing it with “The 3W1H of stakeholder involvement – exploring innovation process”. In that way we believed that presents a better understand of the content.

Point 14:   Lines 177-183: You should refer to an innovation process model that describes different phases or stages of an innovation process; and then refer back to it when you describe your findings in section 3.6.

Response 14: Thank you for your suggestion. We agree that more complex models respond nowadays challenges. In that way, we presented agile stage-gate, proposed by Cooper (2016). Anyway, we are proposing a first connection between RRI and innovation management. In that case, we believe that simpler model “innovation funnel” could clearly demonstrated the inclusion stage.

Point 15:   Line 202: I’d suggest to change the title of section 3 to “Method”, “Approach” or similar and then move lines 84-90 to the beginning of this section.

Response 15: Thank you for your suggestion. We renamed it to “Meta-Synthesis Method”. We also described the guiding goal in the beginning of this section.

Point 16:   Line 203: Describe the goal of the study here.

Response 16: Thank you for your suggestion. We include the description of the goal.

Point 17:   Line 240: To be exact, you should refer to the research on RRI and RI; based on the search terms you described earlier.

Response 17: Thank you for your observation. We included both terms on the sentence.

Point 18:   Line 241: While you searched for articles about “stakeholders”, you use the results to draw a conclusion about “inclusion”. This, in my opinion, is not permissible. I strongly recommend to stick to drawing conclusions about stakeholders; see also my comment #8.

Response 18: Thank you for bringing this to our attention. The term “stakeholders” was used as a search term, as there are several theoretical constructs such as anticipation, inclusion, reflection that all connected to the “stakeholder” term. However, throughout the review, it became clear that inclusion seems to be the key term in RRI. For instance, anticipation and reflection is not possible if inclusion is not in place. That said, we agree that paper would benefit if we use “stakeholder” in a broader perspective then just inclusion. We now have tried to adjust our text to highlight other RRI dimensions that deals with stakeholder involvement. In particular, we reflected don that in introduction and conclusion sections of the paper.

Point 19:   Line 253: Table 1 is a list and doesn’t really present any analysis. How about starting the sentence as “Table 1 presents…”

Response 19: Thank you for calling our attention to this flaw. We followed your suggestion.

Point 20:   Section 3.6: Make sure to word findings in a way that does not suggest that you would generalize your findings. As an example, the sentence in line 367 could start like this: “In the seven case studies that were analysed, the agents who…” or “In the identified sample of case studies…”. Another example; in lines 500-501 you state that “…the contribution of stakeholders does not influence the processes and the position of innovation”; which you cannot certainly not state as a general truth based on your analysis of seven case studies.

Response 20: Thank you for your suggestion. We renamed the sub-section to “Synthesizing stakeholder inclusion in the analysed cases”.

Point 21:   Lines 387-398 offer no answer to the question at which stage of the innovation process stakeholders are included! If you have no information about this, I would suggest removing the question itself from the study. See also my comment #14.

Response 21: Thank you for bringing this to our attention. We made a mistake not describing findings related to that stage. We have now highlighted evidences to answer that question. In this way, we realized that the inclusion is carried out in the final stage of innovation.

Point 22:   Lines 399ff: Here you talk about “the RI literature”. Make sure to stick with the analysis of your seven sample cases in this section.

Response 22: Thank you for making us aware about it. We rewrite some paragraphs in a way to focus our analyses to the seven sample cases. But in a way to advance the discussion, we bring complementary studies. We hope it improve the quality of our study.

Point 23:   Line 438: This might be a language issue, but in my perspective you can draw conclusions from evidence but you cannot not develop, or state, “evidence” based on a literature survey. They could be hypotheses to be further explored in future research.

Response 23: Thank you for that reflection. The goal of a meta-synthesis is to analyze constructs, key variables, and underlying relationships across a set of primary qualitative case studies to arrive at a refined, an extended or even new theory. In this way, considering the findings of the articles, we agree with your comment. In doing "analysis of analysis" (meta-synthesis) we relate a set of propositions, statements about concepts that can be judged as true or false, if they refer to observable phenomena (COOPER and SCHINDLER, 1998).

Point 24:   Section 4: In this section – as well as in the whole paper – you use terms such as “stakeholder”, “agent” and “inclusion” that carry very specific meaning in different academic communitites (such as e.g. the term stakeholder in the CSR literature). You should state clearly if you refer to any specific definition of these terms.

Response 24: Thank you for making us aware about it. We include stakeholder concept and we are guided by it. We did not present a concept to agent, but we present their role. And inclusion was also conceptualized, and we followed that concept.

Point 25:   Conclusion section: Again, make sure to only draw conclusions that are within the limitations of the study method; and phrase them accordingly. Refrain e.g. from talking about “most empirical studies” (line 577).

Response 25: We made a series of improvements in the discussion and conclusion, following your suggestion. In particular, we extended conclusions to include limitations and future research section.

Point 26:   Line 587-588: I do not see how this study is connected to the SDGs. I especially don’t see any contribution to their achievement.

Response 26: Thank you for highlight that. Once we rewrite introduction and conclusion, we expect that the connection between RRI and SDG is clearer. At the same time, as we mentioned in General comments Response 1 and Specific parts of the paper Response 1, we noticed that we are not going to answer the SDG proposal. In that way, we excluded SDG from the title and we just used it to reinforce the importance of stakeholder participation in RRI.

Reviewer 3 Report

General statement.

A very important topic on responsible innovation, possibly the core one. But it will certainly gain on impact on including practices from business and industry, and better differentiating what is Research, what is innovation, and the role of stakeholders for each.

The topic of the article is very relevant to the achievement of SDG’s with societal and environmental impact through responsible innovation. The authors are exploring the ways to address questions about stakeholders such as who is orchestrating their participations, at which stage, with which contribution and who they are, with an attempt to draw conclusions on the basis of selected research papers, mostly from the bio-economy.

The conclusions from this paper are captured in 4 evidences: that the role of stakeholders is motivated by academic researchers, that their role takes place when innovation is already on the market, , involves limited product and paradigm innovations, and about multiple stakeholders.

I see some issues issues with the procedure and the conclusions and I am confident that addressing them would enable the paper to have a much higher impact. I hope that the points below will help the authors for this.

First, the paper does not define enough , and therefore does not differentiate research and innovation. These are very different processes, one being about knowledge generation, the other being about value (mostly economic, but not only) generation. The paper refers to research and innovation, but I would recommend to deep dive into definitions and differences, because we are also facing 2 very different   cultures, academia and business, that do not dialogue enough. The paper could be an opportunity to bridge the gap.

In Innovation Management (I stress: Innovation, and not Research), business is typically applying some tools that should be highlighted for a better authority and credibility of the paper. What is named: anticipation, reflection, deliberation and responsiveness is named differently in the practices of innovation management as applied in business and industry. This may be an explanation why so few academic papers are referring to these practices.

-          First, the innovation management typically follows the approach of stage-gates, with several stages clearly defined. The model of Gary Cooper (google around this for references) is widely applied in one way or another, and it should be referenced, because according to the phase, different practice and requirements on stakeholders involvement would be required.

The authors should really include this knowledge in their work.

-          Second, most innovation is managed through principles of project management, as defined by the PMBOK of the PMI, or the ISO 21500 on project management. Especially in the early phase of project management, the culture of high interaction with stakeholders is captures through the “Agile Project Management” approach, putting emphasis on frequent interactions. How this is fitting with evidence 2 is not clear!

The authors should really include this knowledge in their work.

-          The discussion on stakeholders should be revisited, to reflect practices commonly used in project management. See: the chapter 13th of the PMBOK of the PMI (Project Management Institute).  Stakeholders” should deserve to be better defined, because it encompasses several different types, all with a different role. Additionally, the typology of Ronald Mitchell can bring clarity. And in project management, the concept of stakeholders mapping, and of the prioritization with RACE matrix (responsible, accountable, consulted, informed) is a standard practice that any innovation manager would implement. Not referring to it would make the paper less credible.

-          In innovation project management, the concept of “early management of stakeholders” is also standard: regulatory, quality management, consumers, finance, marketing, etc… This is not visible in the paper.

-          In the design of innovation, the approach of Design Thinking is nowadays a standard practice, and it is applied in some way or another, typically within Agile Project Management. As this relates to consumers involvement, it should be referred to.

-          The impact assessment, such as Environmental or social life cycle assessment, is widely applied, albeit there are some deficiencies here, possibly for discussion within this paper, or separately due to the complexity of the topic.

-          And: the social media is a very important source of reading societal concerns on potential impacts of innovation. While this of course can only apply for innovations already in the market, it is providing a very important source of feedback that is used for future developments or adjustments. The paper would benefit on impact by including this point in the discussion.

On the methodology and the conclusions, the problem here is that the paper is starting from a rather narrow basis of material, with 7 papers, that are used to extrapolate evidences and conclusions that are not really reflecting the experience from industry, again the major driver of innovation. An approach that would rather start from commonly applied best practices, outlined above, and compare them to the learning’s from the 7 papers, would certainly lead to more careful conclusions.

More specifically, reviewing the text:

To: 1. Introduction

There is valuable  approach of linking responsible innovation to the achievements of the SDG’s. Now, the authors seem to be  equating research to innovation. As this is commonly done within the academia (reference 5 of the text), it is a point of view to live with, but this is in fact adding fog to the process, as these are 2 different processes, with 2 very different cultures and even processes.

The inclusion dimension of RI is certainly correct, but the dimension of externalities and E+S impact are not reflected, which is also an important, if not central point to RI. SDG 17 has indeed a dimension of collaboration between agencies, NGO’s and receiving entities, reinforcing the necessity of including stakeholders at an early stage, with emphasis on the assistance of cooperation to development. The authors should however clarify whether they address this point, typically under the government of UN and government bodies, which is normally inclusive, as such type of programs are designed jointly with governmental agencies, or whether they address the process of innovation as an economic/business process.

There are also some statements that are disputable, such as (lines 84-90):

-          The first aspect – the inclusion agent – is not dealt with in the literature

-          We noticed that, although participation is achieved when innovation is already being implemented or has already been introduced to the market (post-launch),

-          The agents who stimulate their participation are academic researchers and researchers linked to multi-institutional projects

Such types of statements are not reflecting the innovation process (or not even the development-aid process in case of close link to SDG’s), or should then be justified.

To 2. The inclusion dimension of responsible innovation

discussion of inclusion is good. It would still be interesting to have a discussion on the limitation of the process, such as:

-          The differentiation for social innovation or product innovation

-          Entrepreneurship is about risk taking, which makes inclusion sometimes difficult, due to the necessity of trade secrets, of potential delays,…

-          Innovation to bring benefits to all stakeholders: not necessarily. Some issues are also about externalities, first difficult to measure and quantify, and that may affect stakeholders differently. And arbitrage can be very controversial.

-          A discussion on transparency could enhance the topic on inclusion.

Small question on line.97:

 is it: which can guarantee sustainable development through a fairer society, achieved through integration

or: which can guarantee a fairer society through sustainable development, achieved through integration

In any case, clarify the meaning.

To 2.1. The stakeholders in RI

Interesting typology of stakeholders. For the rest, see above.

To 2.2. the 3W1H

-          For clarity of the text, better to define acronyms when using them for the first time

-          Line 170: absence of studies on agents responsible for stakeholders inclusions: see above. Risks analysis, project governance and project management practices do care for this!

-          Line 177: some hints on Innovation management best practices, but this should be emphasized, and used in the research.

-          Line 181: none of them explore the theme of the innovation process: does not mean there is no such process, see above!

-          Line 194: point of the selection of stakeholders… who will contribute… This does not reflect practices. Stakeholders are selected on the way they will influence the project, or will be influenced. See above.

To 3. Meta-synthesis

The methodology is well described.

But as innovation is mostly implemented by business, the boundaries of the research could be extended to cover this area as well. A simple google search (not only scopus, or WOS), would do for this.

-          Line 285: the low number of papers (7 articles) should point to an issue: we spend about 3 trillion US$ equivalent yearly on innovation. This cannot happen without relevant practices. The paper should discuss the gap between research on the topic, and the practices!

-          Line 302: consistent with best practices: we are at the cornerstone of the content of the article here. This point should be much better expanded. Just pointing at a reference is by far not enough for the credibility of the paper.

-          Line 312: studes focused on the bioeconomy and health

-          Line 376: the agents stimulating stakeholders participation are academic researchers….: where does this conclusion stem from? As demonstrated by….. is not enough to substantiate such a claim, especially because it will contradict the experience of any innovation practitioner.

-          Line 383: reference to the Alliance project (on sustainable urban interchange) and Pegasus project (on farmland and forest management) should be elaborated. These are research projects, and not innovation projects.

-          Line 390 + fff: interesting discussion on psychology of stakeholders. Could be the topic oif further research.

-          Line 400: see points at the beginning o stakeholders mapping and involvement.

-          Line 415: disputable conclusion. Refer to practices of Design Thinking, a central one in Innovation management

To 4. Discussion.

On evidence 1: the discussion is confusing. On one side, we refer to research, driven by researchers. On the other side, the authors refer to innovation driven by business. 2 very distinctive situations cannot lead to one common conclusion, or evidence, at least not without clearly substantiating it.

-          Line 464: inclusion occurring when product has been developed: clearly not true. Elaborate then how you come to this conclusion, and why it may differ in practice!

-          Line 468: inclusion occurs at the first phase ….: then, elaborate on the discussion, as this is not aligned to evidence 1

-          Line 475+fff: authors refer to research objects, and not stakeholders. Clarify.

On evidence 2: role of stakeholders… when innovation is already on the market! Again, this is conflicting with practices of innovation management and agile project management (design locked-in too early.

-          Line 488-9: good points

-          Line 495: valid point.

On evidence 3: stakeholders…. limits product+ paradigm innovation! The discussion of this paragraph is interesting and well documented. It is also pointing at the necessity to differentiate among the various types of stakeholders, and their potential contribution (see at the beginning on typology and RACI matrix).

On evidence 4: I am not clear on the conclusions here. The paper would gain on impact by elaborating and clarifying this discussion. Mapping of stakeholders is by definition multiple, and some will play a role on RI, others possibly less so. Starting from the typology of stakeholders, and deducting their potential role and contribution could possibly help!

To: 5. Conclusion.

The conclusions are based on the evidences listed above, that are frankly challengeable (see points above).

But the conclusion  clearly emphasizes the importance of stakeholders engagement in order to implement a responsible innovation, and therefore SDG’s.

On this basis, the paper is important, and deserves attention.

But on the other side, it would certainly gain on impact if it would revisit some evidences and conclusions, to get them closer to practices in business and industry.

Line 593: good point, reinforcing the EU approach of mission oriented research and innovation.

Author Response

Dear Reviewer,

We thank you and the other reviewers for the comments and contributions. You have allowed us to considerably improve our paper. We have now uploaded all the documents again via the submission system.

We have endeavoured to attend all the reviewers' recommendations, which have proven to be very useful and coherent. The responses to each of your recommendation or suggestion can be found below.

Before responding to all reviewers’ comments individually, we would like to explain the main changes we made in the paper. These key modifications were motivated by observations that were repeated, in different forms, by all reviewers:

We reorganized the abstract, better      detailing the goal and results of the paper, as our contribution to RRI      literature;

We reorganized the introduction,      seeking to make the paper's contribution clearer;

We insert suggestions of the      evaluators in the literature review, seeking a better positioning for our      contribution;

4.     We reorganized the sub-section Synthesizing stakeholder inclusion in the analysed cases and the discussion of the results, aiming to be more objective;

We reorganized      conclusion, also seeking to make the paper's contribution clearer and      instigating for further research.

Once again, thank you for your fruitful comments, it helped a lot to improve our paper

Best regards,

Authors

General statement.

A very important topic on responsible innovation, possibly the core one. But it will certainly gain on impact on including practices from business and industry, and better differentiating what is Research, what is innovation, and the role of stakeholders for each.

The topic of the article is very relevant to the achievement of SDG’s with societal and environmental impact through responsible innovation. The authors are exploring the ways to address questions about stakeholders such as who is orchestrating their participations, at which stage, with which contribution and who they are, with an attempt to draw conclusions on the basis of selected research papers, mostly from the bio-economy.

The conclusions from this paper are captured in 4 evidences: that the role of stakeholders is motivated by academic researchers, that their role takes place when innovation is already on the market, , involves limited product and paradigm innovations, and about multiple stakeholders.

I see some issues issues with the procedure and the conclusions and I am confident that addressing them would enable the paper to have a much higher impact. I hope that the points below will help the authors for this.

Point 1: First, the paper does not define enough , and therefore does not differentiate research and innovation. These are very different processes, one being about knowledge generation, the other being about value (mostly economic, but not only) generation. The paper refers to research and innovation, but I would recommend to deep dive into definitions and differences, because we are also facing 2 very different   cultures, academia and business, that do not dialogue enough. The paper could be an opportunity to bridge the gap.

Response 1: Thank you for bringing this to our attention. We agree that there is a difference between responsible research and responsible innovation. To fill this gap, we made some improvements on the introduction. There, we let clear that RRI is more related to research while RI is more related to innovation (organizational level). As empirical papers analysed in the paper are based on research-driven innovations often nested in academic environments, we decided to use term RRI consistently throughout a paper, although pointing out that it included also RI. Also, throughout the paper – both in introduction, theory and conclusion, we added literature from other relate domains like innovation management and reflected on the limitation of current research.

Point 2: In Innovation Management (I stress: Innovation, and not Research), business is typically applying some tools that should be highlighted for a better authority and credibility of the paper. What is named: anticipation, reflection, deliberation and responsiveness is named differently in the practices of innovation management as applied in business and industry. This may be an explanation why so few academic papers are referring to these practices.

-          First, the innovation management typically follows the approach of stage-gates, with several stages clearly defined. The model of Gary Cooper (google around this for references) is widely applied in one way or another, and it should be referenced, because according to the phase, different practice and requirements on stakeholders involvement would be required.

The authors should really include this knowledge in their work.

Response 2: Thank you for your suggestion. We totally agree that business tools deal with anticipation, reflection, deliberation and responsiveness. In that sense we rewrite the introduction (reflecting also in the discussion and conclusion), clearing that we aimed to bridge RRI context to innovation management context. We also referenced Cooper’s (2016) agile stage-gate and mentioned briefly other innovation management techniques that are focused on user inclusion.

Point 3:           Second, most innovation is managed through principles of project management, as defined by the PMBOK of the PMI, or the ISO 21500 on project management. Especially in the early phase of project management, the culture of high interaction with stakeholders is captures through the “Agile Project Management” approach, putting emphasis on frequent interactions. How this is fitting with evidence 2 is not clear!

The authors should really include this knowledge in their work.

Response 3: Thank you for your suggestion. Once we tried to better link the connection between RRI and innovation management tools, we referenced PMBook, noticing that is an important tool for business/project management.

Point 4: -          The discussion on stakeholders should be revisited, to reflect practices commonly used in project management. See: the chapter 13th of the PMBOK of the PMI (Project Management Institute).  Stakeholders” should deserve to be better defined, because it encompasses several different types, all with a different role. Additionally, the typology of Ronald Mitchell can bring clarity. And in project management, the concept of stakeholders mapping, and of the prioritization with RACE matrix (responsible, accountable, consulted, informed) is a standard practice that any innovation manager would implement. Not referring to it would make the paper less credible.

-          In innovation project management, the concept of “early management of stakeholders” is also standard: regulatory, quality management, consumers, finance, marketing, etc… This is not visible in the paper.

-          In the design of innovation, the approach of Design Thinking is nowadays a standard practice, and it is applied in some way or another, typically within Agile Project Management. As this relates to consumers involvement, it should be referred to.

-          The impact assessment, such as Environmental or social life cycle assessment, is widely applied, albeit there are some deficiencies here, possibly for discussion within this paper, or separately due to the complexity of the topic.

-          And: the social media is a very important source of reading societal concerns on potential impacts of innovation. While this of course can only apply for innovations already in the market, it is providing a very important source of feedback that is used for future developments or adjustments. The paper would benefit on impact by including this point in the discussion.

Response 4: Thank you for bringing this to our attention. We agree that it is important to conceptualize the stakeholder, what we did at the sub-section 2.1. In that case we used Freeman’s concept. We also agree that in project management, design of innovation, life cycle and social media lies strong value to the innovation process. On the other hand, the objective of the paper is to contribute to RRI literature on the first place. By pointing out (as suggested by reviewer) the importance of bridging RRI and innovation management literature, we hope to add to the development of RRI by raising awareness of the tools and practise available in innovation management domain. However, to keep up the focus of this paper, we decided to pinpoint this “bridging” more broadly, not specifying each tool.  Discussion of all available innovation management techniques we feel is beyond the scope of this paper. We embedded discussion of innovation management domain as related to RRI both in introduction, theory and particular in the conclusion section, pointing to the avenues for future research

Point 5: On the methodology and the conclusions, the problem here is that the paper is starting from a rather narrow basis of material, with 7 papers, that are used to extrapolate evidences and conclusions that are not really reflecting the experience from industry, again the major driver of innovation. An approach that would rather start from commonly applied best practices, outlined above, and compare them to the learning’s from the 7 papers, would certainly lead to more careful conclusions.

Response 5: Thank you for your comments. We agree that were a rather narrow basis of material. It is a consequence of the meta-synthesis method, where the goal is to analyze constructs, key variables, and underlying relationships across a set of primary qualitative case studies to arrive at a refined, an extended or even new theory. In this way, considering the findings of the articles, we agree with your comment. In doing "analysis of analysis" (meta-synthesis) we relate a set of propositions, statements about concepts that can be judged as true or false, if they refer to observable phenomena (COOPER and SCHINDLER, 1998). Since we extend our paper to discuss the possible contribution of insights from innovation management literature to RRI domain, we now reflected on the limitations of our findings in the conclusion section, pointing that our empirical papers under investigation are embedded into academic settings. Thus, findings about type of agents, stage of inclusion osv might be biased. That also leads us to suggesting avenues for future research that should try to take important knowledge from innovation management field into RRI practices.

More specifically, reviewing the text:

Point 1: To: 1. Introduction

There is valuable  approach of linking responsible innovation to the achievements of the SDG’s. Now, the authors seem to be  equating research to innovation. As this is commonly done within the academia (reference 5 of the text), it is a point of view to live with, but this is in fact adding fog to the process, as these are 2 different processes, with 2 very different cultures and even processes.

The inclusion dimension of RI is certainly correct, but the dimension of externalities and E+S impact are not reflected, which is also an important, if not central point to RI. SDG 17 has indeed a dimension of collaboration between agencies, NGO’s and receiving entities, reinforcing the necessity of including stakeholders at an early stage, with emphasis on the assistance of cooperation to development. The authors should however clarify whether they address this point, typically under the government of UN and government bodies, which is normally inclusive, as such type of programs are designed jointly with governmental agencies, or whether they address the process of innovation as an economic/business process.

There are also some statements that are disputable, such as (lines 84-90):

-          The first aspect – the inclusion agent – is not dealt with in the literature

-          We noticed that, although participation is achieved when innovation is already being implemented or has already been introduced to the market (post-launch),

-          The agents who stimulate their participation are academic researchers and researchers linked to multi-institutional projects

Such types of statements are not reflecting the innovation process (or not even the development-aid process in case of close link to SDG’s), or should then be justified.

To 2. The inclusion dimension of responsible innovation

discussion of inclusion is good. It would still be interesting to have a discussion on the limitation of the process, such as:

-          The differentiation for social innovation or product innovation

-          Entrepreneurship is about risk taking, which makes inclusion sometimes difficult, due to the necessity of trade secrets, of potential delays,…

-          Innovation to bring benefits to all stakeholders: not necessarily. Some issues are also about externalities, first difficult to measure and quantify, and that may affect stakeholders differently. And arbitrage can be very controversial.

-          A discussion on transparency could enhance the topic on inclusion.

Response 1: Thank you for your comment. Thank you for making us aware about it. We include stakeholder concept and we are guided by it. We did not present a concept to agent, but we present their role, that we believe is helpful. Inclusion is now more clearly conceptualized in the paper, and we followed that concept.

Point 2: Small question on line.97:

 is it: which can guarantee sustainable development through a fairer society, achieved through integration

or: which can guarantee a fairer society through sustainable development, achieved through integration

In any case, clarify the meaning.

Response 2: Thank you for bringing this to our attention. Although we cannot state this, there are a number of empirical cases that show gains in terms of impact. In that way, we illustrated it using business with the Base-of-Pyramid.

Point 3: To 2.1. The stakeholders in RI

Interesting typology of stakeholders. For the rest, see above.

Response 3: Thank you for your kind comment.

Point 4: To 2.2. the 3W1H

-          For clarity of the text, better to define acronyms when using them for the first time

Response 4: Thank you for bringing this to our attention. We included the definition of the acronyms.

Point 5: Line 170: absence of studies on agents responsible for stakeholders inclusions: see above. Risks analysis, project governance and project management practices do care for this!

Response 5: Thank you for your suggestion. We rewrite introduction and we reinforced that we focused specifically in RRI literature. Unfortunately, RRI literature do not brings the agent. That is the reason that we highlight the gap.

Point 6: Line 177: some hints on Innovation management best practices, but this should be emphasized, and used in the research.

Response 6: Thank you for bringing this to our attention. We made a series of improvements on Introduction, bringing management best practices. But we reinforce that, as mentioned above (General Statement point 4), we bridged it more broadly.

Point 7: Line 181: none of them explore the theme of the innovation process: does not mean there is no such process, see above!

Response 7: Thank you for your comment. We agree that the selected papers do not explore the innovation process, once they are about responsible research (and not innovation). We rewrite introduction and we reinforced that we focused specifically in RRI literature and responsible research. We hope that it’s clearer now.

Point 8: Line 194: point of the selection of stakeholders… who will contribute… This does not reflect practices. Stakeholders are selected on the way they will influence the project or will be influenced. See above.

Response 8: We agree with your comment. We rewrite introduction in a way to let it clearer.

Point 9: To 3. Meta-synthesis

The methodology is well described.

But as innovation is mostly implemented by business, the boundaries of the research could be extended to cover this area as well. A simple google search (not only scopus, or WOS), would do for this.

Response 9: Thank you for positively recognizing our effort to better describe the method. Meta-synthesis consists of an exploratory, inductive research project, based on a systematic review of the bibliography, to synthesize qualitative case studies with the purpose of contributing with detailed action modes and patterns. For this reason, we limited our search on academic database. But we agree that other sources would bring diverse empirical evidences.

Point 10: Line 285: the low number of papers (7 articles) should point to an issue: we spend about 3 trillion US$ equivalent yearly on innovation. This cannot happen without relevant practices. The paper should discuss the gap between research on the topic, and the practices!

Response 10: We agree with your comment. Otherwise we cannot say that it is not happening. Once we focused on RRI literature, we highlight the gap.

Point 11: -          Line 302: consistent with best practices: we are at the cornerstone of the content of the article here. This point should be much better expanded. Just pointing at a reference is by far not enough for the credibility of the paper.

Response 11: Thank you for pointing this out. We agree with that., however, please notice that article refers to supplementary material. The Reading guide supplementary material better described the method used for each case.

Point 12: -          Line 312: studies focused on the bioeconomy and health

Response 12: Thank you for bringing this to our attention. We reorganized the sentence.

Point 13: -          Line 376: the agents stimulating stakeholders participation are academic researchers….: where does this conclusion stem from? As demonstrated by….. is not enough to substantiate such a claim, especially because it will contradict the experience of any innovation practitioner.

Response 13: Your observation helped us to notice that we needed to improve the analyses. In that way, we rewrite the Discussion, pointing that our statement deals in the research level. At the business (organizational) level, it happens in a different way.

Point 14: -          Line 383: reference to the Alliance project (on sustainable urban interchange) and Pegasus project (on farmland and forest management) should be elaborated. These are research projects, and not innovation projects.

Response 14: Thank you for your observation. We rewrite the Discussion, reinforcing that our meta-synthesis worked with research projects, not innovation ones.

Point 15: -          Line 390 + fff: interesting discussion on psychology of stakeholders. Could be the topic oif further research.

Response 15: Thank you for your suggestion. We included it for further studies.

Point 16: -          Line 400: see points at the beginning o stakeholders mapping and involvement.

Response 16: Thank you for your comment. We rewrite discussion and conclusion, and we reinforced that we focused specifically in RRI literature and the analyses is based on research projects.

Point 17: -          Line 415: disputable conclusion. Refer to practices of Design Thinking, a central one in Innovation management

Response 17: Thank you for your suggestion. We rewrite discussion and conclusion, and we reinforced that we focused the analyses based on research projects.

Point 18: To 4. Discussion.

On evidence 1: the discussion is confusing. On one side, we refer to research, driven by researchers. On the other side, the authors refer to innovation driven by business. 2 very distinctive situations cannot lead to one common conclusion, or evidence, at least not without clearly substantiating it.

Response 18: Thank you for this point, and also to your overall reflections about academic and business worlds. As discussed above, we have now extended dour paper to include discussion on innovation management and practices coming from organizational levels as potential tolls and practices that can be applied also in the case of academic-based innovations (research projects). We rewrote the Discussion section to make this clearer.

Point 19: -          Line 464: inclusion occurring when product has been developed: clearly not true. Elaborate then how you come to this conclusion, and why it may differ in practice!

Response 19: Thank you and again we now highlighted the specific context of the empirical studies analysed – academic settings. We reflected in both Discussion and Conclusion of that limitation. We rewrote the discussion to make this clearer.

Point 20: -          Line 468: inclusion occurs at the first phase ….: then, elaborate on the discussion, as this is not aligned to evidence 1

Response 20: We rewrote the discussion to make this clearer. In that way, we reinforced that we are describing the academic contexts.

Point 21: -          Line 475+fff: authors refer to research objects, and not stakeholders. Clarify.

Response 21: Thank you for your observation. We rewrote the sentence bringing the benefits to stakeholders, excluding the objects.

Point 22: On evidence 2: role of stakeholders… when innovation is already on the market! Again, this is conflicting with practices of innovation management and agile project management (design locked-in too early.

Response 22: Thank you for bringing this to our attention. We rewrote introduction and discussion to make this clearer.

Point 23: -          Line 488-9: good points

Response 23: Thank you for your kind comment.

Point 24: -          Line 495: valid point.

Response 24: Thank you for your kind comment.

Point 25: On evidence 3: stakeholders…. limits product+ paradigm innovation! The discussion of this paragraph is interesting and well documented. It is also pointing at the necessity to differentiate among the various types of stakeholders, and their potential contribution (see at the beginning on typology and RACI matrix).

Response 25: Thank you for your comment. We rewrote introduction and discussion to make this clearer.

Point 26: On evidence 4: I am not clear on the conclusions here. The paper would gain on impact by elaborating and clarifying this discussion. Mapping of stakeholders is by definition multiple, and some will play a role on RI, others possibly less so. Starting from the typology of stakeholders, and deducting their potential role and contribution could possibly help!

Response 26: Your question made us think about the contribution of evidence 4. We rewrote the evidence and the discussion the discussion related to it.

Point 27: To: 5. Conclusion.

The conclusions are based on the evidences listed above, that are frankly challengeable (see points above).

But the conclusion  clearly emphasizes the importance of stakeholders engagement in order to implement a responsible innovation, and therefore SDG’s.

On this basis, the paper is important, and deserves attention.

But on the other side, it would certainly gain on impact if it would revisit some evidences and conclusions, to get them closer to practices in business and industry.

Response 27: Thank you for making us aware about it. We rewrote the conclusion to make this clearer.

Point 28: Line 593: good point, reinforcing the EU approach of mission oriented research and innovation.

Response 28: Thank you for your kind comment.

Round 2

Reviewer 1 Report

To be honest, I still have some reservations: To me, it feels like the authors put the goal of 'doing a meta-lit-review' above the goal of 'answering the research question'. What I mean with this: Citing Hoon (2013) they (repeatedly) stress that they are doing a meta-lit-review, according to the rules, so to say--but they seem to put relatively less effort in finding other ways to answer the research questions; which they could have done by looking at or even only obliquely, superficially hinting at 'grey' literature.

Author Response

Dear Reviewer,

Thank you for your comments and for the opportunity to improve our paper. Your suggestions made a great difference in the way we looked at the paper.

Best regards,

Authors

Point 1: To be honest, I still have some reservations: To me, it feels like the authors put the goal of 'doing a meta-lit review' above the goal of 'answering the research question'. What I mean with this: Citing Hoon (2013) they (repeatedly) stress that they are doing a meta-lit-review, according to the rules, so to say--but they seem to put relatively less effort in finding other ways to answer the research questions; which they could have done by looking at or even only obliquely, superficially hinting at 'grey' literature.

Response 1: Thank you for calling our attention to this.  Maybe we have cited Hoon (2013) – repeatedly -  because this is a new methodology that promises us to promote conclusions using practical, context-dependent (Stake, 2005; Yin, 2009) knowledge from case studies.  When we access meta-analysis (statistical technique specially developed to integrate the results of two or more independent studies), we do not normally question whether the rules were followed, since the precepts of such methods come from the seventeenth century. However, since metasynthesis are still very recent in management studies, we repeatedly use the citation feature as a way of validating the choices made. In this context it is important to highlight that it was never our intention to do a literature review, but rather a metasynthesis, which is a methodology that provided us with higher quality data than any other type of literature could offer (especially because grey literature not always follow  standards of scientific rigor and data aggregation or interpretation, that we can find in case studies peer reviewed and accepted for publishing in journals).

The use of case studies has its strength in producing new theoretical knowledge stemming from contextualized case-specific findings (Eisenhardt, 1989).  Once metasynthesis use only qualitative case studies, using it, we were able to explore how people and organizations give meaning to the world around them, seeking to interpret the meanings brought by individuals about the multiple phenomena. In other words, to answer our research question that was linked to the “role played by stakeholders,” this meaning could only be captured in qualitative case studies. Thus, the proposed answer to the question about the role played by the stakeholders uses variables that go beyond the sum of the parts, since they offer a new interpretation of the results. The new interpretation cannot be found in any primary research report since they are inferences derived from the fact that all articles have become a sample as a whole (Sandelowski et al., 1997; Sandelowski & Barroso, 2003; Hoon, 2013).

Although we have deepened only 7 articles, we do not fail to evaluate in depth a great number of articles, exactly so that the selected articles could be used to reach the objective of the article. In this sense, we have developed a table with all 217 valid articles originally selected. In this table were applied filters with the criteria established for exclusion. Because it is a large volume of articles, we chose not to make it available as additional material, as it would require translation. The original article and all its complementary material were written in Portuguese.

Seeking to contribute to our arguments, we attach the original table (in Portuguese, now – to respect the deadline from the editor, but we can translate if we receive some days more). In addition, we highlight and illustrate some exclusion criteria for articles. In the following paragraphs, we seek to demonstrate the research criteria and exclusion criteria adopted beyond the analysis of the papers.

We found 55 case studies. As we mentioned in the article (line 382 and follow), 15 articles with illustrative cases were excluded, because they didn´t demonstrate or explored the meanings given to the stakeholder. For example, despite the alignment and relevance of articles such as Bakker et al. (2014), that was in this set of 55 articles, which focused on issues of power, information (asymmetries) and responsibility, describing Dutch policy, assessments and debates on nanotechnology in general and on nanofood in particular. Specifically, this study was not used because it did not bring the stakeholder into the discussion, just mentioned its importance, at the same time the reluctance to share or communicate information with stakeholders because of the fear that the discussions take a 'wrong direction.' Other articles such as those of Saleeby (2014) and Macnaghten (2016), have the same characteristic.

In the same way, 15 articles were excluded in which the analysis does not correspond to the responsible innovation process or does not consider stakeholders, were excluded. We excluded articles such as Fitzgerald et al. (2016), which emphasize the importance of citizen participation as a key component in policy-making. The authors present a predominant question as to the extent to which citizens' participation in decision making makes a difference in the formulation of policies for technology. However, the article proposes to present a methodology for its inclusion, not advancing in including and reporting the result with this inclusion. In addition, 7 articles on cases based exclusively on documents (reports or policies), like Owen & Goldberg (2010) e Stemerding (2015). Were also excluded 8 case studies that use quantitative methods for the data analysis, like Foley, et al. (2016), 2 articles that are not available in the databases (Halme, & Korpela, 2013; Arniani, 2016) and 1 that is a case of an RRI teaching method (Marschalek, et al., 2017).

Once more we would like to thank you for helping us to stress the use of the method, that which we believe to be appropriate for discussions such as the one we have proposed, but which requires even greater dissemination and reinforcement of the results obtained.

Reference

Arniani, M. (2015, October). Technology, Citizens and Social Change in the Framework of European Research and Innovation Programmes: Towards a Paradigm Shift. In International Internet of Things Summit (pp. 233-238). Springer, Cham.

de Bakker, E., de Lauwere, C., Hoes, A. C., & Beekman, V. (2014). Responsible research and innovation in miniature: Information asymmetries hindering a more inclusive ‘nanofood’development. Science and Public Policy41(3), 294-305.

Eisenhardt, K. M. (1989). Building theories from case study research. Academy ofManagementReview, 14, 532-550.

Fitzgerald, C., McCarthy, S., Carton, F., Connor, Y. O., Lynch, L., & Adam, F. (2016). Citizen participation in decision-making: can one make a difference?. Journal of Decision systems25(sup1), 248-260.

Halme, M., & Korpela, M. (2013). Scarcity or abundance? Examination of resources behind responsible innovation in small enterprises. In Academy of Management Proceedings(Vol. 2013, No. 1, p. 12877). Briarcliff Manor, NY 10510: Academy of Management.

Hoon, C. (2013). Meta-synthesis of qualitative case studies: An approach to theory building. Organizational Research Methods16(4), 522-556.

Macnaghten, P. (2016). Responsible innovation and the reshaping of existing technological trajectories: the hard case of genetically modified crops. Journal of Responsible Innovation3(3), 282-289.

Marschalek, I., Schrammel, M., Unterfrauner, E., & Hofer, M. (2017). Interactive reflection trainings on RRI for multiple stakeholder groups. Journal of Responsible Innovation4(2), 295-311.

Owen, R., & Goldberg, N. (2010). Responsible innovation: a pilot study with the UK Engineering and Physical Sciences Research Council. Risk analysis: An international journal30(11), 1699-1707.

Saleeby, E., Holschneider, C. H., & Singhal, R. (2014). Paradigm shifts: using a participatory leadership process to redesign health systems. Current Opinion in Obstetrics and Gynecology26(6), 516-522.

Sandelowski, M., Docherty, S., & Emden, C. (1997). Qualitative metasynthesis: Issues and techniques. Research in nursing & health20(4), 365-371.

Sandelowski, M., & Barroso, J. (2003). Creating metasummaries of qualitative findings. Nursing research52(4), 226-233.

Stake, R. E. (2005). Qualitative case studies. In N. K. Denzin & Y. S. Lincoln (Eds.), The Sage handbook of qualitative research (3rd ed., pp. 443-465). Thousand Oaks, CA: Sage.

Teece,

Stemerding, D. (2015). iGEM as laboratory in responsible research and innovation. Journal of Responsible Innovation2(1), 140-142.

Yin, R. (2009). Case study research: Design and methods (3rd ed.). Thousand Oaks, CA: Sage.

Reviewer 3 Report

This is a fundamentally revised version of the fist text, and while it is reviewing Research on RI, it is now covering as well practices of innovation and project management. This makes the paper much more relevant in my view. As such, it is worth to be published.

Some minor comments below.

To 1: introduction. The text was extended, and is relating now to practices that are not for whatever Reason included in academic Research. 

To 2: Responsible Innovation: text extended.

- interesting comments on CSR

- interesting clarifications to the role of stakeholders in §2.1

- L 270: about several forms of innovations: Don't forget innovations on business models, how a product or service is brought to consumers, same for societal innovation.

- L 280: selection of stakeholders: those who contribute, but also those who are impacted. 

To 3. Meta-synthesis: minor revisions, text OK

To 4: Discussion: interesting attempt to bridge Research on RRI and innovation management.

- around proposition 2: in a next study, you could possibly relate the constraints of the role of stakeholders to intellectual property issues. Co-creation and open innovation are not really coexisting well with the requirements for patenting. And liabilities related to  risk management can set limitations to stakeholders involvment.

- around proposition 4, and external stakeholders: to make things simple, the potential of nuisance of a stakeholder will greatly influence on the way it is included. A minority cluster in preference mapping will be included very differently than a regulator, or a NGO with a high level of expertize…. Could help for further Research!

To 5: Conclusions. Indeed, the easons for a late inclusion of external stakeholders could be the topic of a furhter Research, together with the impact on innovation

Author Response

Dear Reviewer,

Thank you for your comments and for the opportunity to improve our paper. Your suggestions made a great difference in the way we looked at the paper.

Best regards,

Authors

Point 1: Comments and Suggestions for Authors

This is a fundamentally revised version of the fist text, and while it is reviewing Research on RI, it is now covering as well practices of innovation and project management. This makes the paper much more relevant in my view. As such, it is worth to be published.

Response 1: Thank you for your kind comment regarding our effort in revising the article.

Point 2: To 1: introduction. The text was extended, and is relating now to practices that are not for whatever Reason included in academic Research. 

Response 2: Thank you for positively recognizing our effort to relate these practices to the RRI context.

Point 3: To 2: Responsible Innovation: text extended.

- interesting comments on CSR

- interesting clarifications to the role of stakeholders in §2.1

Response 3: Thank you for your kind comment.

Point 4: - L 270: about several forms of innovations: Don't forget innovations on business models, how a product or service is brought to consumers, same for societal innovation.

Response 4: Thank you for your observation. We agree that there are different kinds of innovation, as you pointed, and we were focus on product/service development. In that sense, we included as a suggestion for further studies the analyses of the inclusion on the development of new business models, societal innovations, among others.

Point 5: - L 280: selection of stakeholders: those who contribute, but also those who are impacted. 

Response 5: Thank you for bringing it for our attention. We rewrite the sentence.

Point 6: To 3. Meta-synthesis: minor revisions, text OK

Response 6: Thank you for your kind comment.

Point 7: To 4: Discussion: interesting attempt to bridge Research on RRI and innovation management.

Response 7: Thank you for positively recognizing our effort to bridge RRI and innovation management.

Point 8: - around proposition 2: in a next study, you could possibly relate the constraints of the role of stakeholders to intellectual property issues. Co-creation and open innovation are not really coexisting well with the requirements for patenting. And liabilities related to risk management can set limitations to stakeholders involvment.

Response 8: Thank you for highlighting these points. We agree that there is a dichotomy between secrecy and transparency, which generates a selective opening of information. This is due to the fear of reducing competitive advantage to competitors, loss of control over the product and fear of knowledge leakage. The consequence of transparency is a lower level of patents (Spinello, 2003), mainly at the first steps of the innovation process, regardless the patent costs (Blok, 2015).  In this sense, there is scope to be explored in future works and we add that suggestion.

Point 9: - around proposition 4, and external stakeholders: to make things simple, the potential of nuisance of a stakeholder will greatly influence on the way it is included. A minority cluster in preference mapping will be included very differently than a regulator, or a NGO with a high level of expertize…. Could help for further Research!

Response 9: Thank you for highlighting it. We included the suggestion for future studies.

Point 10: To 5: Conclusions. Indeed, the easons for a late inclusion of external stakeholders could be the topic of a furhter Research, together with the impact on innovation

Response 10: Thank you for your attention. We already suggested investigating the reason for late inclusion (Line 696), but we rewrite the sentence to include the impact on innovation.
